# SPECS: DECOUPLING MULTIMODAL LEARNING VIA SELF-DISTILLED PREFERENCE-BASED COLD START

**Kun Chen**[1,2,*], **Peng Shi**[3,*], **Haibo Qiu**[3], **Zhixiong Zeng**[2,3], **Siqi Yang**[3], **Wenji Mao**[2,1,†],
**Lin Ma**[3,†]

[1]School of Artificial Intelligence, University of Chinese Academy of Sciences
[2]MAIS, Institute of Automation, Chinese Academy of Sciences
[3]Meituan
{chenkun2024, wenji.mao}@ia.ac.cn, shipeng10@meituan.com,
forest.linma@gmail.com

## ABSTRACT

Reinforcement learning with verifiable rewards (RLVR) has recently catalyzed a wave of "MLLM-r1" approaches that bring RL to vision language models. Most representative paradigms begin with a cold start, typically employing supervised fine-tuning (SFT), to initialize the policy before RL. However, SFT-based cold start adopts the reasoning paradigm intertwined with task solution and output format, which may induce instruction-style overfitting, weakens out-of-distribution generalization, and ultimately affects downstream RL. We revisit the cold start along two views, its training method and data construction, and introduce the *Generalization Factor* (GF) coefficient to quantify the generalization capability under different methods. Our empirical study finds that preference–based training methods (e.g. DPO) generalizes better than SFT-based methods in cold start. Motivated by this, we propose **SPECS**—a **S**elf-distilled, **Pr**eference-based **C**old **S**tart framework that decouples multimodal learning: (1) generates introspective preference data pairs via self-distillation, avoiding reliance on larger teachers or manual annotation; (2) performs preference–based training to learn, focusing on shallow, transferable surface-form criteria (format, structure, style) rather than memorizing content; and (3) hands off to RLVR for deep reasoning results. Experimental results across multiple multimodal benchmarks show that our decoupling learning framework yields consistent performance gains over strong baselines, improving MEGA-BENCH by 4.1% and MATHVISTA by 12.2%. Additional experiments indicate that SPECS contributes to reducing in-distribution "stuckness," improving exploration, stabilizing training, and raising the performance ceiling.

Project Page: https://kwen-chen.github.io/SPECS-VL/

## 1 INTRODUCTION

Recently, inspired by the success of DeepSeek-R1 (Guo et al., 2025), in effectively enhancing the reasoning capabilities of large models through reinforcement learning (RL) with verifiable reward (Lambert et al., 2024; Guo et al., 2025), a growing body of work has begun to apply RL directly to vision language models (VLMs). This has led to a wave of exciting "MLLM-r1" research (Meng et al., 2025; Shen et al., 2025; Peng et al., 2025; Zhou et al., 2025; Zhang et al., 2025b; Wang et al., 2025b;a; Zheng et al., 2025; Ma et al., 2025; Lan et al., 2025; Qiu et al., 2025), which leverage similar principles to advance multimodal reasoning.

Previous research has indicated that prior to RL, employing a pre-training or warm-up phase (which is termed **"cold start"**), can significantly improve the readability, stability, and even the final performance of RL training (Guo et al., 2025). Currently, the most commonly used cold start strategy is supervised fine-tuning (SFT) , where the model is first fine-tuned on a set of high-quality reasoning data to provide a better initial policy for the subsequent RL phase (Wei et al., 2025; Yang et al.,

---

*Equal contribution. †Corresponding authors.

2025b; Huang et al., 2025; Deng et al., 2025b). This strategy enables the model to be trained on complex reasoning data during the cold start phase, thereby acquiring reasoning ability.

The common understanding behind SFT-based cold start is that reasoning abilities, reasoning format and other learning objectives can be jointly learned during the cold start phase. However, such an SFT-based joint learning paradigm may largely affect the model's generalization capability (Wu et al., 2025; Chu et al., 2025), and consequently degrade subsequent RL (Chen et al., 2025a). This raises an important research issue of quantifying and improving the model's *generalization capability* during cold start and working in concert with subsequent RL.

To address the above limitations, we consider an alternative learning paradigm, which separates the learning process into hierarchical stages based on the idea that cold start phase focused more on shallow learning to avoid prematurely getting stuck in in-distribution problem solving, while subsequent RL focuses on the deep-level learning of a solution to boost the overall performance (Bengio et al., 2009). Thus, the intuition of our adopting decoupling learning for multimodal reasoning is that the selection of pre-training methods in cold start needs to better support the subsequent RL, both in terms of generalization and by having separate objectives to facilitate better final results.

Another important issue is the generation of cold start data. Previously, the prohibitive cost of human annotation has motivated a growing body of research to explore the use of synthetic data. This often involves using a more capable large model as a "teacher" to distill data for a smaller "student" model. (Zhang et al., 2025c; Yao et al., 2024; Xu et al., 2024; Huang et al., 2025). However, when the capability gap between the teacher model and the student model is too large, it can lead to a decline in model performance (Zhang et al., 2023). Alternatively, the DeepSeek-R1-Zero paradigm (Guo et al., 2025) first directly applies RL to the base model for obtaining R1-Zero and then generates cold start data by zero model itself. This paradigm has achieved very remarkable performance; yet it still has the limitation of reliance on the SFT cold start and the constraints between SFT and subsequent RL, thereby leaving room for further improvement.

In this paper, to examine the suitable cold start training method, we propose the *Generalization Factor* (GF) coefficient in Section 2 to quantify the generalization capability of the model and conduct an empirical study to evaluate different training methods. We identify that Direct Preference Optimization (DPO) (Rafailov et al., 2023) based on preference data is a cold start approach that enables the model to have better performance. On this basis, we present the **S**elf-distilled **Pr**eference-based **C**old-**S**tart (SPECS) framework in Section 3. By decoupling the learning objectives during DPO to focus on output format, we create a pre-aligned model that serves as a superior starting point for the final RL fine-tuning. Our experiments show that this method leads to more stable, efficient training, and a higher performance ceiling compared to the advanced and strong baseline.

The main contributions of this paper can be summarized as follows.

1. We present the **SPECS** framework, a three-stage cold start strategy. It generates preference data through self-distillation, uses DPO for cold start training, and separates training objectives so that the model first aligns with output formats, providing a stronger starting point for RL.

2. We propose **Generalization Factor** as a metric to evaluate a model's generalization capability under different cold start training methods by comparing its performance on in-distribution and out-of-distribution tasks.

3. We reveal the importance of **Decoupling Learning** between the cold-start and RL phases. This separation improves exploration and reduces the risk of the model getting stuck on in-distribution solutions.

4. Our experiments prove that a DPO cold start gives the model stronger generalization ability. In terms of the model's final results, it achieves consistent performance gains across benchmarks, improving MEGA-Bench by 4.1% and MathVista by 12.2% over strong baselines.

## 2 EMPIRICAL INVESTIGATION

### 2.1 EVALUATING DEGREE OF GENERALIZATION

To evaluate the impact of preference-based versus supervised data on a model's generalization capabilities under a fixed sample size, we introduce the metric of **Generalization Factor (GF)**.

**Setup**. We define an evaluation function $\psi(f_n, P) \in \mathbb{R}$ that measures the performance of a model $f$ on a data distribution $P$ and $n$ refers to the size of the training data samples. A higher value of $\psi$ indicates better performance.

**Generalization Factor**. To accurately evaluate the generalization ability of a model, we first need to test the model's performance on in-distribution (ID) and out-of-distribution (OOD) tasks. Among them, ID tasks require the same as the task requirements during training, while OOD tasks require different from the task requirements during training.

- ID Performance: $\Psi_{\text{ID}}(n)$, is evaluated on a hold-out set from the same distribution $P_{train}$.

$$\Psi_{\text{ID}}(n) = \psi(f_n, P_{train})$$

- OOD Performance: $\Psi_{\text{OOD}}(n)$, is the weighted average performance across a set of $m$ distinct OOD distributions, $Q = \{Q_1, \ldots, Q_m\}$, with weights defined by a distribution $\alpha$.

$$\Psi_{\text{OOD}}(n) = \mathbb{E}_{Q \sim \alpha}[\psi(f_n, Q)]$$

We establish a baseline model, $f_0$, which serves as a reference point. The performance gains over this baseline are calculated as:

$$G_{\text{ID}}(n) = \Psi_{\text{ID}}(n) - \Psi_{\text{ID}}(0)$$

$$G_{\text{OOD}}(n) = \Psi_{\text{OOD}}(n) - \Psi_{\text{OOD}}(0)$$

We define GF, $\Gamma(n)$ as the $F_\beta$-score of the model with respect to OOD performance gains and ID performance gains. The reason for adopting this metric is that the $F_\beta$-score is particularly suitable for average ratios. Its most prominent feature is that the result tends to lean toward the smaller number. This perfectly aligns with our needs: as long as either the ID or OOD performance is very poor, the final score will be very low. We can also control the size of $\beta$ to reflect the degree of importance we attach to OOD performance gains during the training process.

$$\Gamma(n) = (1 + \beta^2)\frac{G_{\text{ID}}(n)G_{\text{OOD}}(n)}{\beta^2 \cdot G_{\text{ID}}(n) + G_{\text{OOD}}(n)}$$

where the weighting coefficient $\beta$ is generally set to 2 to reflect the importance of the OOD performance gain in the generalization capabilities of the model. To ensure that the metric behaves well and is dimensionless, the evaluation function $\psi$ should be normalized to a consistent range.

## 2.2 EXPERIMENTAL FINDINGS

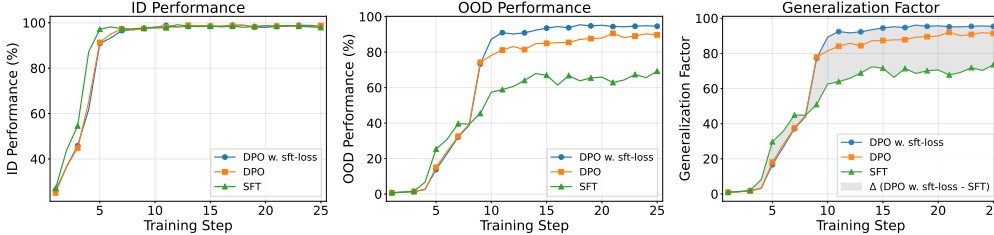

Figure 1: Performance Comparison: DPO vs. SFT on In-Distribution and Out-of-Distribution Task

To preliminarily examine how preference data and supervised data affect model generalization, we construct a preference dataset $\mathcal{D}_{\text{pref}} = \{(x_i, y_i^+, y_i^-)\}_{i=1}^N$ , where $y_i^+$ is the chosen response and $y_i^-$ is the rejected response, and a supervised dataset $\mathcal{D}_{\text{SFT}} = \{(x_i, y_i)\}_{i=1}^N$ with $y_i = y_i^+$ around reasoning tasks defined by a specific answer format. Under equal data budgets, we evaluate two settings: (i) an in-distribution setting in which the required reasoning format matches that used in training, and (ii) an out-of-distribution setting in which the required reasoning format differs [1]. We compare DPO training, SFT training, and DPO training augmented with SFT loss (see Section 3.3). The resulting $\Psi_{\text{ID}}(n)$, $\Psi_{\text{OOD}}(n)$, and the $\Gamma(n)$ are reported in Figure 1.

---

[1]For more detailed description, see Appendix B

From the experimental results, it can be observed that SFT achieves the fastest convergence on ID tasks. However, due to its reliance on a single cross-entropy loss that maximizes the log-likelihood of the correct answer, it demonstrates poor OOD performance. By contrast, DPO converges more slowly at the beginning of ID tasks but yields better OOD performance. Remarkably, the model trained with a combination of DPO with SFT loss achieves the strongest generalization capability overall. As the number of training steps increases, the GF gap between the SFT training method and the DPO training method also increases.

# 3 METHODOLOGY: THE SPECS FRAMEWORK

## 3.1 SELF-DISTILLED PREFERENCE COLD-START

A model with superior generalization capabilities provides a more effective starting point for RL. Inspired by the discussion in Section 2, we employ self-distillation to construct preference data focusing format learning. This data is then used in place of standard SFT data to enhance the model's generalization performance during the cold-start phase.

To implement this, we propose SPECS, illustrated in Figure 2, a three-stage training optimization strategy consisting of **1) Self-Distillation for Preference Data Generation**, **2) DPO-based Pre-Alignment for Cold-Start**, and **3) Final GRPO Fine-tuning**.

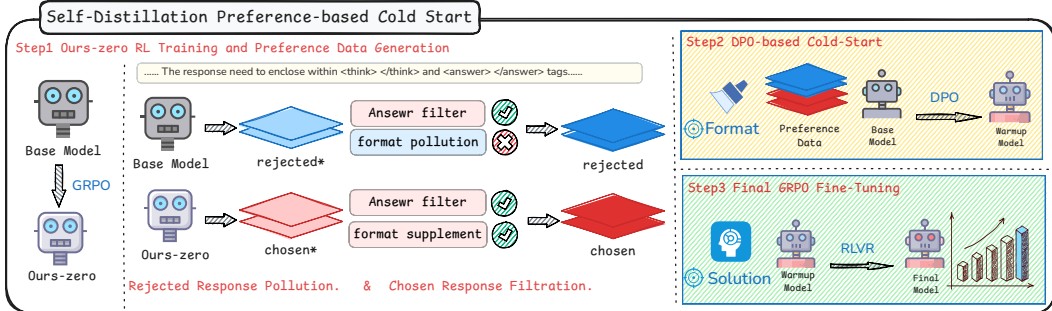

Figure 2: **Method Overview.** We propose the SPECS cold-start strategy, a three-stage pipeline to enhance final RL fine-tuning. Firstly, where we generate a preference dataset focused on teaching the correct output format by self distillation. Next, The base model is pre-aligned on this data using DPO to create a format-aware "Warmup Model". Finally, this pre-aligned model undergoes Final RL tuning with GRPO, allowing the optimization process to focus on enhancing reasoning.

## 3.2 SELF-DISTILLATION FOR PREFERENCE DATA GENERATION

**Objective:** The foundational stage of our framework aims to achieve two interconnected goals: first, to cultivate a preliminary "seed model" with enhanced reasoning capabilities, and second, to leverage this model to autonomously generate a high-quality preference dataset through a process we term self-distillation.

**Methodology:** A critical initial challenge is that a standard base VLM often lacks the capability to generate outputs of sufficient reasoning ability. To address this, we first conduct a brief, initial phase of RL fine-tuning on the base model using GRPO. This step aims not at achieving the final performance, but at creating an initial policy, denoted $\pi_{GRPO-zero}$, which is more adept at exploring the solution space.

With the exploratory $\pi_{GRPO-zero}$ model, we proceed to generate the preference dataset. The data construction process involves four key steps:

- **Response Generation.** We prompt two models, our exploratory $\pi_{GRPO-zero}$ and $\pi_{base}$, with specific format instructions ( `<think>...</think><answer>...</answer>` ) to create a dataset, which is designed to contain pairs of responses that are both correct in their final answer, but differ in their reasoning paradigm and answer format.

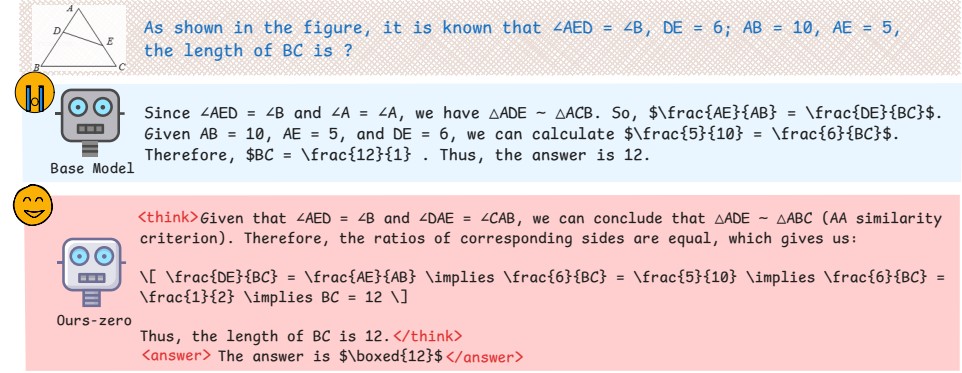

Figure 3: Example of a self-distilled preference data pair.

- **Chosen Response Filtration.** For the chosen response ($y_i^+$), we use Gemini-2.5 flash (Comanici et al., 2025) as an evaluator. Assesses whether the reasoning path in the $\pi_{GRPO-zero}$ response aligns correctly with its final answer. Only responses in which the reasoning and the answer are consistent are retained, forming a high-quality pool of candidates. For more analysis of this content, please refer to Appendix E.

- **Rejected Response Pollution.** For the rejected response ($y_i^-$), we select responses that also contain the correct answer, but deviate from the required format. Recognizing that some generated responses might incidentally have the correct format, We randomly apply one of the following five types of format corruption to these responses to ensure a clear learning signal.

  1. Remove all tags (`<think>`, `</think>`, `<answer>`, `</answer>`).
  2. Remove the `<answer>` and `</answer>` tags.
  3. Remove the `<think>` and `</think>` tags.
  4. Remove the `<answer>` and `</answer>` tags and move the closing `</think>` tag to the end of the response.
  5. Replace the `<answer>` tags with the string `Answer:` and remove `</answer>` tags.

- **Preference Pair Construction via Self-Distillation.** We construct the chosen response and the rejected response into pairs of self-distilled preference data $(y^+, y^-)$. As shown in Figure 3. Both Chosen Responses ($y_i^+$) and Rejected Responses ($y_i^-$) are selected from the filtered pool and contain the correct final answer. This data set is designed to facilitate decoupled learning, separating the learning of reasoning paradigms and answer formats from the core logical reasoning ability. This approach serves as a more effective cold-start method for the final alignment stage.

## 3.3 DPO-BASED PRE-ALIGNMENT FOR COLD-START

**Objective:** The primary goal of this stage is to leverage the self-distilled preference dataset generated in the Stage 1 (Section 3.2) to pre-align the base VLM. This process yields a "cold-start" model that serves as a significantly improved starting point for the final RL fine-tuning. We conceptualize this phase as a "warm-up," which shifts the model's policy into a more advantageous region of the policy landscape before the intensive final training.

**Methodology:** To achieve this pre-alignment, we employ DPO (Rafailov et al., 2023), a powerful technique that directly optimizes the language model on preference data without the need for an explicit reward model. The standard DPO loss function is defined as:

$$\mathcal{L}_{DPO}(\pi_\theta; \pi_{\text{ref}}) = -\mathbb{E}_{(x,y_w,y_l)\sim D} \left[ \log \sigma \left( \beta \log \frac{\pi_\theta(y_w|x)}{\pi_{\text{ref}}(y_w|x)} - \beta \log \frac{\pi_\theta(y_l|x)}{\pi_{\text{ref}}(y_l|x)} \right) \right]$$

where $\pi_\theta$ is the policy being optimized, $\pi_{ref}$ is the reference policy (the initial base model), $\beta$ is a temperature parameter, and $(x, y_w, y_l)$ represents a triplet of prompt, chosen response, and rejected response from our self-distilled dataset D.

To augment this process, we incorporate an SFT loss computed on the "chosen" samples, which serves as a form of regularization. It ensures that while the model learns the directional preference signal from DPO, it does not drift far from the core distribution of high-quality text embodied by the chosen responses (Rao et al., 2025). The combined loss function is thus:

$$\mathcal{L}_{hybrid} = \mathcal{L}_{DPO} + \lambda \mathcal{L}_{SFT}$$

where $\mathcal{L}_{SFT}$ is the conventional negative log-likelihood loss on the chosen responses, and $\lambda$ is a weighting coefficient to balance the two objectives. For a discussion for $\lambda$, see Appendix C.

### 3.4 FINAL GRPO FINE-TUNING

**Objective:** To achieve peak performance by fine-tuning the pre-aligned cold-start model, focusing computational resources on enhancing complex reasoning capabilities.

**Methodology:** This final stage leverages the cold-start model obtained from Stage 2 as the initialization point for RL, rather than starting from the base model or a conventional SFT model. The pre-alignment from the DPO phase ensures that the model has already mastered the output format. Consequently, the model is not required to expend resources on learning basic structural compliance. Instead, credit assignment during RL training can be more accurately attributed to the core challenge: improving the quality and precision of its reasoning process. This targeted optimization explains the observed stable convergence in our experiments and the model's ability to achieve a higher performance ceiling.

For the final stage of fine-tuning, we employ the GRPO algorithm (Shao et al., 2024). This process is guided by a composite reward function that combines format and accuracy components to evaluate the model's output, $o$, for a given question, $q$.

The total reward $R_{total}$, is the sum of a format reward $R_{format}$, and an accuracy reward $R_{acc}$:

$$R_{total}(o, q) = R_{format}(o) + R_{acc}(o, q)$$

**The format reward $\mathbf{R_{format}(o)}$**, assigns a fixed value of 0.5 for structurally correct outputs, reinforcing the policy's formatting discipline.

**The accuracy reward $\mathbf{R_{acc}(o, q)}$**, provides a binary signal: 1.0 for a correct answer and 0 otherwise. We use a hybrid mechanism to determine correctness based on the question type, $T(q)$:

$$R_{acc}(o, q) = \begin{cases} R_{\text{rule}}(o, q) & \text{if } T(q) \in \{\text{Multiple-Choice, Numerical}\} \\ R_{\text{llm}}(o, q) & \text{if } T(q) = \text{Short-Answer} \end{cases}$$

For objective types like multiple-choice and numerical questions, a rule-based function assesses correctness. For subjective short-answer questions, we employ GPT-4o as an external judge.

## 4 EXPERIMENTS

### 4.1 EXPERIMENT SETTINGS

**Dataset and Benchmark:** The data utilized for training $\pi_{GRPO-zero}$ model in Stage 1 and for the final GRPO fine-tuning of the cold-started model in Stage 3 is composed of the Orsta47K (Ma et al., 2025) and virl39K (Wang et al., 2025a) datasets. In Stage 2 of cold start training, we used 9K self-distilled data. This composition is designed to enhance the model's general and mathematical reasoning capabilities. We conduct evaluations on multiple benchmark datasets, including MEGA-Bench (Chen et al., 2025b), MMMU (Yue et al., 2024), MathVista (Lu et al., 2024b), MATH-Vision (Wang et al., 2024a), and MathVerse (Zhang et al., 2024).

**Baseline:** Our comparative analysis is grounded on two primary categories of models. The first category comprises open-source general VLMs, including QwenVL-2-7B (Wang et al., 2024b), QwenVL-2.5-7B (Bai et al., 2025), InternVL2-8B (Chen et al., 2024), InternVL2.5-8B (Chen et al., 2024), Kimi-VL-A3B (Team et al., 2025), and DeepSeek-VL-7B (Lu et al., 2024a). The second category focuses on models specifically engineered for advanced reasoning tasks. This group includes Kimi-VL-A3B-Thinking (Team et al., 2025), R1-Onevision (Yang et al., 2025b), VLAA-Thinking

Table 1: Model performance comparison on MEGA-Bench Core.

| Model | MEGA-Bench | | | | | | | | MEGA-Bench |
|---|---|---|---|---|---|---|---|---|---|
| | Knowledge | Mathematics | Perception | Coding | Info. Ex. | Planning | Science | Metrics | Core |
| *Open-Source General Models* | | | | | | | | | |
| QwenVL-2-7B | 39.96 | 25.95 | 39.99 | 31.49 | 40.29 | 16.64 | 28.59 | 43.61 | 34.47 |
| QwenVL-2.5-7B | 38.84 | 27.67 | 41.24 | 28.93 | 50.23 | 16.32 | 36.75 | 41.64 | 35.07 |
| InternVL2-8B | 33.94 | 22.08 | 32.15 | 24.7 | 29.13 | 12.17 | 24.61 | 39.96 | 25.96 |
| InternVL2.5-8B | 34.78 | 25.86 | 33.27 | 25.45 | 35.10 | 15.97 | 28.83 | 44.96 | 28.34 |
| InternVL3-8B | **42.76** | **34.85** | 42.76 | 34.05 | 44.84 | 17.10 | 35.21 | 49.60 | 36.02 |
| Llava-OV-7B | 31.37 | 22.11 | 27.64 | 13.9 | 17.07 | 9.16 | 24.38 | 37.31 | 21.36 |
| Kimi-VL-A3B | 37.63 | 27.07 | 39.50 | 22.30 | 40.99 | **22.17** | 33.94 | 46.65 | 34.40 |
| *Open-Source Reasoning Models* | | | | | | | | | |
| R1-Onevision† | 29.47 | 20.94 | 28.65 | 23.38 | 43.04 | 12.67 | 26.84 | 42.19 | 27.18 |
| VLAA-Thinking† | 38.23 | 28.83 | 40.73 | 28.84 | 44.58 | 17.05 | 36.69 | 45.57 | 34.86 |
| Kimi-VL-A3B-Thinking | 33.45 | 17.76 | 28.11 | 14.69 | 41.14 | 12.64 | 28.60 | 43.97 | 27.08 |
| MM-Eureka-7B | 40.12 | 31.59 | 39.71 | 28.75 | 49.32 | 16.64 | 37.25 | 46.39 | 35.96 |
| VL-Rethinker-7B | 40.65 | 30.08 | 42.02 | 29.87 | 52.03 | 17.83 | 36.82 | 46.90 | 37.25 |
| Orsta-7B | 41.65 | 31.48 | 43.84 | 32.82 | **54.07** | 17.83 | 36.91 | 41.66 | 38.31 |
| Ours-zero | 42.44 | 29.87 | 43.77 | 32.80 | 49.59 | 17.76 | 37.39 | 47.32 | 37.96 |
| **Ours-7B** | 42.64 | 31.71 | **44.58** | **34.14** | 51.68 | 18.76 | **38.73** | **51.87** | **39.17** |
| Δ (Ours - Backbone) | +3.8 | +4.0 | +3.3 | +5.2 | +1.4 | +2.4 | +2.0 | +10.2 | +4.1 |

[1] The †symbol indicates that the results were evaluated with VLMEvalKit[2].
[2] The remaining results are from the MEGA-Bench Leaderboard and Ma et al. (2025).

Table 2: Model Performance Comparison On Other Benchmarks

| Model | MMMU val | MathVision | MathVisita | MathVerse vision only | Overall |
|---|---|---|---|---|---|
| *Backbone* | | | | | |
| QwenVL-2.5-7B | 54.2† | 25.40 | 63.70 | 38.20 | 45.38 |
| *QwenVL-2.5-7B based Reasoning Models* | | | | | |
| R1-Onevision | 49.67† | **29.90** | 64.1 | 40.0 | 45.92 |
| VLAA-Thinking | 52.67† | 26.40 | 68.00 | 48.20 | 48.82 |
| MM-Eureka-7B | 55.55† | 26.90 | 73.00 | 47.58† | 50.76 |
| VL-Rethinker-7B | 56.7 | 29.70 | 73.60 | **48.98**† | 52.25 |
| Orsta-7B† | 54.33 | 25.76 | 70.20 | 32.10 | 45.60 |
| Ours-zero | 54.3 | 26.88 | 72.90 | 47.33 | 50.35 |
| **Ours-7B** | **56.78** | 29.50 | **75.90** | 48.73 | **52.73** |
| Δ (Ours - Backbone) | +2.5 | +4.1 | +12.2 | +10.5 | +7.3 |

The †symbol indicates that the results were evaluated with VLMEvalKit[3].

(Chen et al., 2025a), MM-Eureka-7B (Meng et al., 2025), VL-Rethinker-7B (Wang et al., 2025a), and Orsta-7B (Ma et al., 2025).

**Implementation Details:** We utilize the open-source Multimodal Large Language Model, Qwen2.5-VL-7B (Bai et al., 2025), as our base model. For the GRPO training in Stage 1 and Stage 3, we employ the MM-EUREKA [4] framework. The training batch sizes are both set to 128, with 8 rollouts generated per sample. For the DPO training in Stage 2, as well as for the comparative SFT experiments, we leverage the LlamaFactory [5] framework. In this configuration, the training batch size is set to 64, and the hyperparameter $\lambda$ for the hybrid loss function is set to 1. The prompt used during training is shown in Appendix B. Some more detailed settings can be found in Appendix F.

## 4.2 MAIN RESULTS

Table 1 presents the overall performance of our model on MEGA-Bench Core, in comparison with other baseline models. Table 2 reports the performance of various inference models built on the QwenVL-2.5-7B backbone across additional benchmarks. Our model has improvements in general task benchmarks (MEGA-BENCH core, MMMU) and mathematical reasoning benchmarks (Math-Vision, MathVisita, MathVerse), and some benchmarks are in a leading position among models of the same size, demonstrating the effectiveness of our approach.

---

[4] https://github.com/ModalMinds/MM-EUREKA
[5] https://github.com/hiyouga/LLaMA-Factory

### 4.3 ABLATION ON SELF-DISTILLATION AND DECOUPLED DATA STRATEGY

**Self-distillation proves more effective than external teacher models.** First, we evaluate the effectiveness of the self-distillation mechanism by substituting it with preference data generated from powerful external teacher models, specifically QwenVL-2.5-32B and QwenVL-2.5-72B. The results shown in Tabel 4 clearly indicate that our our approach outperforms both teacher-based alternatives. We also observe that performance degradation is more pronounced when using the QwenVL-2.5-32B model, whose output distribution diverges more significantly from base model. This finding suggests that preference data closely aligned with the model's intrinsic capability distribution is more effective for alignment than guidance from a more capable but dissimilar external model.

**Distilling from GRPO-zero instead of the base model.** We directly perform RL on the base model to obtain GRPO-zero, and then distill the chosen response through the GRPO-zero model. This choice of scheme is based on considerations of data utilization and training data quality. As shown in Table 3 below, we have conducted statistics on some indicators of the responses of the original model and the GRPO-zero model to the training data questions. Obviously, the GRPO-zero model has higher data utilization in the collection of chosen responses due to its higher format accuracy and answer accuracy. At the same time, we counted the number of reasoning words (including transition words, causal words, sequential words, etc.) per 1000 characters for both models. We also used the same data collection method to collect responses with correct formatting and correct answers distilled from the base model as chosen responses. The experimental results shown in Table 4. **The**

Table 3: Statistical indicators of response in training data for the base model and GRPO-Zero model

| Model | Format Acc. (%) | Answer Acc. (%) | Reasoning Words / 1k Chars |
|---|---|---|---|
| Qwen2.5-7B-Instruct | 41.62 | 30.42 | 4.26 |
| Ours-GRPO-zero | 96.74 | 52.82 | 4.99 |

**decoupled data strategy outperforms the coupled approach for DPO cold-starting.** Next, we investigate the impact of our decoupled data strategy for DPO cold-starting. We compare it against a "coupled" DPO approach, where preference data is mixed, containing pairs that differ in both answer correctness and reasoning format. The experimental results shown in Table 4 demonstrate the clear superiority of the decoupled approach. We found that while coupled data helps initially, decoupled data provides a better foundation for the main RL phase. We attribute this to decoupled data's sharp focus: it trains only the reasoning paradigms during the cold start, which ultimately leads to better performance after RL, even if the initial cold-start performance is lower.

Table 4: Ablation Results to show the impact of Self Distillation and Decoupled Data

| Model | Megabench | MMMU | MathVista | MathVision | MathVerse | AVG |
|---|---|---|---|---|---|---|
| Qwen-VL-2.5-7B | 35.07 | 54.2 | 63.70 | 25.40 | 38.20 | 43.31 |
| - Qwen32b Distillation | 27.04 / 29.87 | 51.44 / 56.67 | 66.90 / 71.50 | 25.53 / 28.03 | 43.53 / 46.07 | 42.89 / 46.43 |
| - Qwen72b Distillation | 34.00 / 37.30 | 53.89 / 58.56 | 67.50 / 73.30 | 25.62 / 28.91 | 43.53 / 46.83 | 44.90 / 48.98 |
| - Base model Distillation | 35.37 / 37.92 | 53.11 / 56.11 | 67.90 / 74.40 | 25.55 / 28.68 | 43.40 / 46.82 | 45.07 / 48.79 |
| - *Self Distillation* | 37.52 / 39.17 | 54.89 / 56.78 | 72.00 / 75.90 | 25.75 / 29.50 | 46.19 / 48.73 | 47.27 / 50.02 |
| - Coupled Data | 37.02 / 38.76 | 55.44 / 55.44 | 71.10 / 73.10 | 27.37 / 28.65 | 47.46 / 47.46 | 47.67 / 48.68 |
| - *Decoupled Data* | 37.52 / 39.17 | 54.89 / 56.78 | 72.00 / 75.90 | 25.75 / 29.50 | 46.19 / 48.73 | 47.27 / 50.02 |

The value on the left side of the slash '/' represents the score after cold-start training, and the value on the right side of the slash represents the score after cold start + RL.

### 4.4 ANALYSIS OF THE IMPACT OF DPO-BASED COLD START AND SFT-BASED COLD START

We examine the downstream effects of our DPO cold-start strategy, assessing its impact on the efficiency and stability of the final RL phase.

**Performance and Training Efficiency.** To evaluate performance and training efficiency, we tracked MEGA-Bench scores throughout the GRPO training process. As illustrated in Figure 4, the DPO-based GRPO model begins with a substantially higher initial score, demonstrating the immediate benefit of preference-based pre-alignment. Furthermore, it maintains a clear advantage throughout

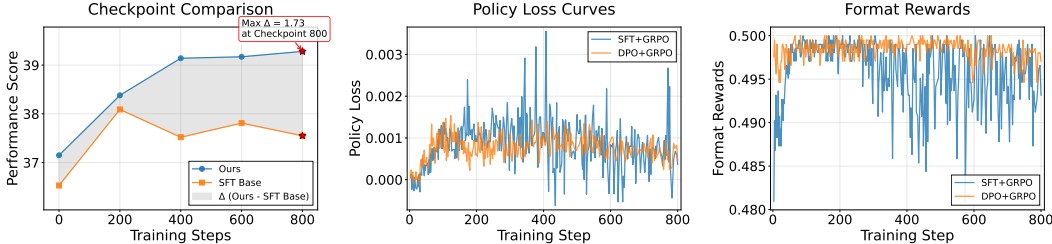

Figure 4: Impact on RL Training Efficiency and stability.

training, converging more rapidly and ultimately achieving a higher performance ceiling than its SFT-based GRPO counterpart. The final performance comparison of other benchmarks is shown in Table 5 below. For more analysis on this content, please refer to Appendix D.

Table 5: Performance Comparison of SFT-based, and DPO-based Models on Various Benchmarks

| Model | Megabench | MMMU | MathVista | MathVision | MathVerse | AVG |
|---|---|---|---|---|---|---|
| Qwen2.5-7B-Instruct | 35.07 | 54.20 | 63.70 | 25.40 | 38.20 | 43.31 |
| SFT-based GRPO | 37.52 | 54.44 | 74.10 | 28.61 | 43.60 | 47.65 |
| DPO-based GRPO | 39.17 | 56.78 | 75.90 | 29.50 | 48.73 | 50.02 |

**Training Stability.** Beyond performance metrics, we analyzed training stability by comparing the policy loss curves, presented in Figure 4. The curve for DPO-based GRPO is visibly smoother and more stable, indicating a more consistent and reliable optimization trajectory. In contrast, the SFT-based GRPO policy exhibits greater volatility, suggesting that the RL algorithm make more drastic and potentially erratic updates. In terms of format rewards, RL based on SFT cold start is also weaker than RL based on DPO cold start in terms of the stability of format rewards. Regarding the impact of different cold-start training methods on the stability of model training, we believe this is related to the training objectives of the cold-start phase and the RL phase. The SFT training objective is to maximize log likelihood, which is a form of imitation learning, while the loss function of DPO can be seen as directly optimizing an implicit reward model consistent with preference data, which is more aligned with the subsequent reward-driven GRPO optimization objective. Therefore, using a DPO-based model as a starting point also brings more stable training for subsequent RL.

### 4.5 ANALYSIS OF THE RELATIONSHIP BETWEEN GF AND FINAL PERFORMANCE

We evaluate the correlation between the model's GF value during the cold-start phase and its final performance (represented here by the average score on MEGA-Bench, MMMU, and MathVerse_Vision_Only). By comparing three cold-start methods with different GF values, presented in Figure 5, we can see that GF and the model's final performance are correlated to a certain extent. This also confirms that the stronger the model's generalization ability in the cold-start phase, the more it will contribute to the model's improvement in the RL phase.

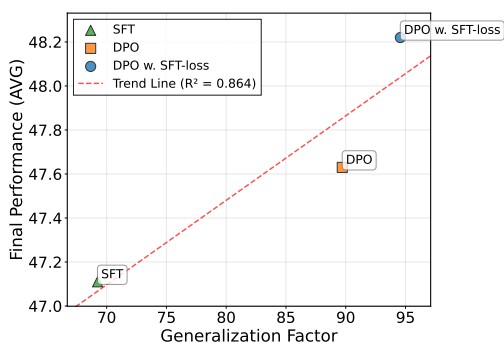

Figure 5: GF vs. Final Performance

In addition, we can also prove that the cold-start method based on preference data has higher generalization ability compared with the traditional SFT cold-start paradigm, thus bringing greater potential for improvement to subsequent RL.

## 5 RELATED WORK

The application of RL has emerged as a highly effective method for enhancing the reasoning capabilities of large language models, with notable successes in the text-only domain such as DeepSeek-R1 (Guo et al., 2025), which leverages RLVR (Lambert et al., 2024; Guo et al., 2025). Inspired by these advancements, a substantial and rapidly growing body of research has begun adapting RL techniques for VLMs. This has catalyzed a wave of "MLLM-r1" studies, all aiming to harness similar principles to unlock more advanced multimodal reasoning abilities. For instance, **MM-Eureka** (Meng et al., 2025) explores the enhancement of multimodal reasoning abilities through rule-based RL by constructing high-quality multimodal reasoning datasets. **VL-Rethinker** (Wang et al., 2025a) stimulates the slow thinking and self-reflection abilities of VLMs through RL. **Orsta** (Ma et al., 2025) establishes a unified RL system that supports VLMs in jointly learning visual reasoning and perception tasks. **VLM-R1** (Shen et al., 2025) extends R1-style RL to VLMs for visual understanding tasks to improve their visual reasoning abilities. **LMM-R1** (Peng et al., 2025) enhances the model's basic reasoning ability and multimodal generalization ability through a two-stage training strategy of basic reasoning enhancement and multimodal generalization training. **R1-VL** (Zhang et al., 2025b) realizes the self-improvement of MLLMs' reasoning ability by solving the sparse reward problem through Step-wise GRPO. **DeepEyes** (Zheng et al., 2025) motivates the model's "Thinking with Images" ability through RL. **PEARL**(Zhang et al., 2025a) strengthens multimodal reasoning by explicitly anchoring it to verified visual evidence. **VisualThinker-R1-Zero** (Zhou et al., 2025) performs RL directly without any supervised fine-tuning of the model to reproduce the "aha moment".

A crucial precursor to effective RL is the "cold-start" phase, which initializes the model's policy before the RL stage begins. The conventional strategy for this phase is SFT, a foundational step adopted by many leading models to establish a strong baseline performance (Wei et al., 2025; Yang et al., 2025b; Huang et al., 2025; Deng et al., 2025b). In parallel with refining cold-start methods, the prohibitive cost of human annotation has driven the field towards synthetic data generation. This approach often involves using powerful teacher models to distill vast amounts of data for training smaller student models (Zhang et al., 2025c; Xu et al., 2024; Huang et al., 2025). **Vision-R1** (Huang et al., 2025) cold-starts the model before applying RL by synthesizing 100K high-quality long CoT instructions. **LLaVA-CoT** (Xu et al., 2024) integrates multiple mainstream visual question answering datasets and uses advanced large models to synthesize 99K valid image-question-answer pairs. Yang et al. (2025a) explicitly decouple the two abilities of "abstract reasoning" and "strategy awareness" through two curriculum-style stages: SFT cold start and RL. **R1-Onevision** (Yang et al., 2025b) adopts a two-stage training strategy of SFT + RL by synthesizing a 155K instruction set.

## 6 CONCLUSIONS

In this study, we introduced the Self-Distilled Preference-based Cold-Start framework, a novel three-stage methodology. By leveraging a self-distillation process to generate preference data, we decouple the learning of shallow objectives, such as output format, from the deep, logical reasoning skills targeted during the final RL phase. Our method utilizes DPO to pre-align the model, providing a superior initial policy for RL. The creative insight of decoupling learning objectives solves the practical problem of SFT-induced overfitting, which often constrains exploration and leads to suboptimal performance. Our results demonstrate the practical value of this approach. The introduction of the Generalization Factor also provides a valuable new metric for quantifying model generalization. This work shows considerable application prospects for developing more robust and capable multimodal reasoning systems.

Despite these promising results, this study has certain limitations that suggest avenues for future research. Our experiments were focused on the multimodal domain; further studies should be conducted to validate the efficacy of the SPECS framework in text-only reasoning tasks. The generalization of our findings could also be strengthened through more extensive testing across a more diverse set of out-of-distribution benchmarks. Such investigations would continue to refine our understanding of how to most effectively structure learning pipelines for complex AI systems.

ACKNOWLEDGEMENT

This work is supported in part by the National Natural Science Foundation of China under Grant #72293575, and the Joint Research Project on the Integration of Culture, Science and Technology between Chinese Academy of Sciences and Hunan Province #2024JK4003.

ETHICS STATEMENT

All of the paper's authors have read and adhered to the ICLR Code of Ethics. This work focuses on advancing the reasoning capabilities of multimodal large language models through a novel training methodology. The core contributions are algorithmic, aimed at improving the efficiency, stability, and performance ceiling of reinforcement learning pipelines for such models.

**Data and Models:** The datasets used for training and evaluation, including Orsta47K, virl39K, MEGA-Bench, MMMU, MathVista, MATH-Vision, and MathVerse, are publicly available datasets and benchmarks established within the academic community. Our use of these standard datasets is intended to ensure transparency, facilitate reproducibility, and allow for direct comparison with prior work. We do not use any private or sensitive user data. The base model used, Qwen2.5-VL-7B, is an open-source model, promoting accessibility and further research.

**External Evaluators:** For evaluating subjective short-answer questions where automated rule-based metrics are insufficient, we employed proprietary models (GPT-4o) as external judges. We acknowledge that these models may have their own inherent biases. This approach was chosen to provide a consistent and scalable evaluation standard for complex, open-ended responses, a common practice in current AI research. The specific prompts and evaluation criteria were designed to be as objective as possible to mitigate these potential biases.

**Potential Societal Impact:** The goal of this research is to enhance the general reasoning abilities of AI systems. While this can lead to positive applications in fields like education, scientific research, and accessibility tools, we recognize that, like any powerful technology, it could potentially be misused. Our work does not introduce any new applications but rather improves the underlying training methodology. We encourage the responsible development and deployment of AI systems built upon these foundational research advancements.

**Bias and Fairness:** Our proposed framework, SPECS, is not designed for a specific downstream application and was evaluated on broad-domain academic benchmarks. We have not conducted an in-depth analysis of social or demographic biases, as the datasets primarily consist of math, science, and general knowledge problems. We acknowledge that the underlying base model and training data may contain biases, and future work should investigate how different training strategies impact the propagation or mitigation of such biases.

REPRODUCIBILITY STATEMENT

To ensure the reproducibility of our work, we provide a detailed account of our methodology and experimental setup. The core SPECS framework, including its three stages of self-distillation for data generation, DPO-based pre-alignment, and final GRPO fine-tuning, is described in Section 3. Our complete experimental settings, including the datasets, benchmarks, and baselines used for evaluation, are detailed in Section 4.1 . This section also specifies crucial implementation details, such as learning rates and batch sizes for all training stages . We utilized publicly available frameworks, MM-EUREKA and LlamaFactory, for our implementation. Furthermore, the appendix offers additional resources to aid in reproduction, including the exact system prompts used for training and inference (Appendix B) and a detailed analysis of the hybrid loss coefficient (Appendix C).

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

## A    THE USE OF LARGE LANGUAGE MODELS

In preparing this manuscript, we utilized the Large Language Model (LLM) Gemini-2.5-pro (Comanici et al., 2025) to polishing the text. Its application was strictly limited to correcting spelling and grammatical errors. The authors manually reviewed and verified all AI-assisted modifications to ensure factual accuracy. The core ideas, methodologies, and figures presented are entirely the original work of the human authors.

## B    PROMPTS

The following are the System Prompts for the model during the cold start training phase and the RL training phase. In the instruction format generalization experiment in Section 2, the Inference System Prompt for the ID Task is consistent with these.

---

**System Prompt for Cold Start Training and ID Task Inference**

```
Solve the question. The user asks a question, and you solve it. You
first think about the reasoning process in the mind and then provide
the user with the answer. The answer is in latex format and wrapped
in $...$. The final answer must be wrapped using the \boxed{}
command. The reasoning process and answer are enclosed within <think>
 </think> and <answer> </answer> tags, respectively, i.e., <think>
Since $1+1=2$, so the answer is $2$. </think> <answer> The answer is
$\boxed{2}$ </answer> , which means assistant's output should start
with <think> and end with </answer>.
```

---

The following is the Inference System Prompt for the OOD Task in the instruction format generalization experiment of Section 2.

---

**System Prompt for OOD Task Inference**

```
Solve the question. The user asks a question, and you solve it. You
first think about the reasoning process in the mind and then provide
the user with the answer. The answer is in latex format and wrapped
in $...$. The final answer must be wrapped using the \boxed{}
command. The reasoning process and answer are enclosed within <cot>
</cot> and <response> </response> tags, respectively, i.e., <cot>
Since $1+1=2$, so the answer is $2$. </cot> <response> The answer is
$\boxed{2}$ </response> , which means assistant's output should
start with <cot> and end with </response>.
```

---

The following Prompt is used for filtering chosen responses, see Section 3.2.

---

**System Prompt for Chosen Response Filtration**

```
You are an expert evaluator. Please analyze the following
problem-solving response and evaluate whether the final answer is
consistent with the reasoning steps.
Question: {question}
Response to analyze: {chosen_answer}
Please output only digit 1 if the answer is consistent with the
reasoning, or digit 0 if the reasoning has obviously theoretical
errors OR the answer is inconsistent with the reasoning.
```

---

## C ANALYSIS OF LOSS BALANCE COEFFICIENTS IN COLD-START TRAINING

### C.1 PRELIMINARIES

**Supervised Fine-Tuning.** SFT adapts pre-trained models by optimizing a cross-entropy loss function to maximize the log-likelihood of a desired output $y_c$ for a given prompt $x$. The loss is defined as:

$$\mathcal{L}_{SFT}(\pi_\theta) = \mathbb{E}_{(x,y_c)\sim D}\left[-\log \pi_\theta\left(y_c|x\right)\right]$$

By training exclusively on positive examples, SFT creates a sharply peaked probability distribution that closely mimics the training data. While this accelerates model convergence, it can limit generalization capabilities.

**Direct Preference Optimization.** DPO(Rafailov et al., 2023) refines models by learning directly from preference data, consisting of a prompt $x$, a chosen response $y_c$, and a rejected response $y_r$. The DPO loss function is designed to increase the relative probability of the chosen response over the rejected one:

$$\mathcal{L}_{DPO}(\pi_\theta;\pi_{\text{ref}}) = -\mathbb{E}_{(x,y_c,y_r)\sim D}\left[\log \sigma\left(\beta \log \frac{\pi_\theta(y_c|x)}{\pi_{\text{ref}}(y_c|x)} - \beta \log \frac{\pi_\theta(y_r|x)}{\pi_{\text{ref}}(y_r|x)}\right)\right]$$

where $\pi_\theta$ is the policy being optimized, $\pi_{\text{ref}}$ is a reference policy, $\sigma$ is the logistic function and $\beta$ is a temperature parameter. This approach maximizes the margin between chosen and rejected responses, cultivating a smoother and more robust probability distribution that enhances the model's generalization performance.

### C.2 ANALYSIS OF LOSS BALANCE COEFFICIENTS

In Section 3.3, we proposed that in our cold start, in addition to the basic DPO loss, we also added the SFT loss to ensure that the model does not deviate too much from the chosen samples during the training process. The combined loss function is thus:

$$\mathcal{L}_{hybrid} = \mathcal{L}_{DPO} + \lambda\mathcal{L}_{SFT}$$

Among the $\mathcal{L}_{hybrid}$, the DPO loss function is designed to increase the relative probability of the chosen response over the rejected one:

$$\mathcal{L}_{hybrid} = -\mathbb{E}_{(x,y_c,y_r)\sim D}\left[\log \sigma\left(\beta \log \frac{\pi_\theta(y_c|x)}{\pi_{\text{ref}}(y_c|x)} - \beta \log \frac{\pi_\theta(y_r|x)}{\pi_{\text{ref}}(y_r|x)}\right)\right] + \lambda\mathbb{E}_{(x,y_c)\sim D}[-\log \pi_\theta(y_c|x)]$$

We recorded the model's rewards for chosen, rewards for rejected, and margins during cold start training when $\lambda = 0$, $\lambda = 0.5$, and $\lambda = 1$, as shown in Figure 6 below.

From the picture, we can see a lot. Since the training loss of DPO can expand the margin between the chosen and the rejected, when $\lambda = 0$, this margin is the largest. However, this margin causes both the chosen and rejected rewards to decrease (the rejected rewards decrease faster than the chosen rewards). When $\lambda = 0.5$ and $\lambda = 1$, it ensures that the chosen rewards increase during the DPO training process.

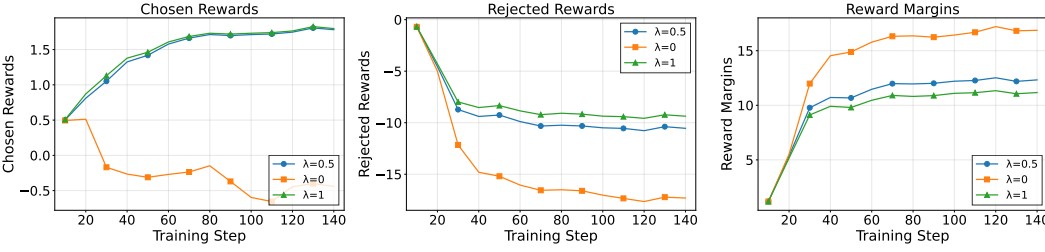

Figure 6: Effect of $\lambda$ on Chosen and Rejected Rewards and Reward Margins During Training

Table 6: Average RBF by Sample Size

| Model / Sample Size | 120 | 240 | 480 |
|---|---|---|---|
| base | 1.7686 | 1.9166 | 1.9206 |
| sft | 1.8086 | 1.8961 | 1.9336 |
| ours | **1.8216** | **1.9268** | **1.9596** |

## D  PERFORMANCE POTENTIAL OF DPO-BASED COLD-START VS. SFT

To evaluate the effectiveness of our proposed cold-start strategy, we first compare our DPO-based approach against a conventional SFT baseline. For this comparison, the SFT model was fine-tuned exclusively on the "chosen" responses from our preference dataset, while our model was trained using the full preference pairs with the DPO algorithm.

**Performance Potential via Pass@K.** We assessed the performance of both the DPO and SFT cold-start models on the MMMU benchmark. As delineated in Figure 7, the DPO-based model demonstrates superior performance across evaluated metrics, including Pass@8, and Pass@32. This indicates that the DPO cold-start method endows the model with greater initial capabilities and higher potential for future alignment tasks compared to the standard SFT approach.

**Exploration Capability via Rollout Branching Factor (RBF).** Beyond task performance, we investigated the intrinsic exploratory capacity of models, a critical factor for successful RL. We measure this using the RBF (Deng et al., 2025a), which quantifies the diversity of a model's generation by counting the

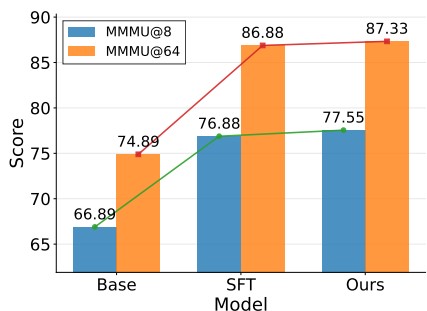

Figure 7: MMMU Pass@K Performance

number of candidate tokens within the probability mass of top p (0.95) during decoding. A higher RBF signifies greater generation diversity and, consequently, a stronger capacity for exploration. As shown in Table 6, our DPO-based cold-start method yields a substantially higher RBF than the SFT baseline. This finding suggests that our approach cultivates a broader exploration space, which is highly beneficial for the subsequent RL phase, enabling the model to discover more diverse and potentially higher-quality solutions. This outcome highlights a key advantage of our method in preparing models for alignment.

## E  ANALYSIS OF CHOSEN RESPONSE FILTRATION

We believe it is crucial to ensure the quality and accuracy of chosen response in cold start parse. However, we found that there are a small number of cases (generated by GRPO-zero) in chosen response where the answers are not consistent with the reasoning, which could affect training, such as the following case:

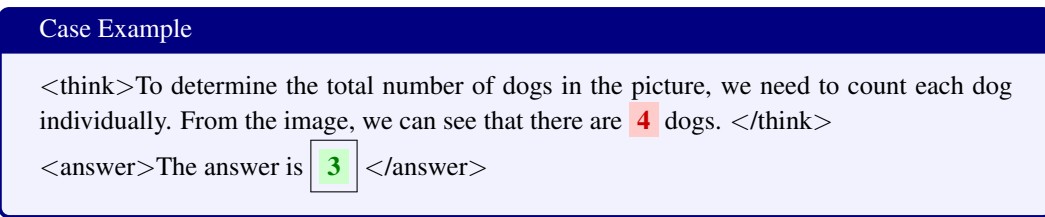

The reasons for this situation still need to be explored, and we think it may be related to reward hacking in GRPO training (because the answer to this training data was incorrectly labeled as 3, while the correct answer should be 4 as in the reasoning process). To prevent these data from affecting the cold start training, we use another LLM to verify the data. This evaluation process is simple and objective, with no significant errors, and will not introduce bias into data filtering. The

table 7 below shows the differences in the final model performance between the data filtered by the advanced LLM and the randomly filtered data.

Table 7: Performance Comparison of Self-Distillation with and without Verification on Various Benchmarks

| Model | Megabench | MMMU | MathVista | MathVision | MathVerse | AVG |
|---|---|---|---|---|---|---|
| Qwen2.5-7B-Instruct | 35.07 | 54.20 | 63.70 | 25.40 | 38.20 | 43.31 |
| w.o verification | 38.72 | 55.11 | 73.50 | 26.90 | 46.44 | 48.13 |
| w verification | 39.17 | 56.78 | 75.90 | 29.50 | 48.73 | 50.02 |

It can be seen from the experimental results that the model after data filtering performs better than the final model obtained by randomly selecting data. On the other hand, regarding the selection of large models for screening, in addition to using some advanced large models, due to the simplicity of the task instructions, some open-source large models can also be used as alternatives.

## F    EXPERIMENTS IMPLEMENTATION DETAILS

All experiments were conducted on a cluster of 32 NVIDIA H800 (80G) GPUs. We utilize the open-source Multimodal Large Language Model, Qwen2.5-VL-7B (Bai et al., 2025), as our base model.

For the GRPO training in Stage 1 and Stage 3, we employ the MM-EUREKA [6] framework. The rollout and training batch sizes are both set to $128$, with $8$ rollouts generated per sample. The learning rate is configured to $1 \times 10^{-6}$. Following the DAPO (Yu et al., 2025) methodology, we set the clipping thresholds to $0.2$ (lower) and $0.28$ (upper), and we do not apply a KL penalty. The maximum output length is restricted to $10,240$ tokens. This training phase spanned $400$ steps and was completed in approximately $12$ hours.

For the DPO training in Stage 2, as well as for the comparative SFT experiments, we leverage the LlamaFactory [7] framework using a total dataset size of $9,155$ samples. In this configuration, the training batch size is set to $64$, the learning rate is maintained at $1 \times 10^{-6}$, and the hyperparameter $\lambda$ for the hybrid loss function is set to $1$. The maximum output length is extended to $16,384$ tokens. These models were trained for $140$ steps, with the training process completing in under $30$ minutes. The prompt used during training is shown in Appendix B.

We have counted the computing time for DPO and SFT training using 8 GPUs under the same training parameters (including steps, batch size, data length, max new tokens, etc.), as shown in the following table 8.

Table 8: Comparison of training performance between SFT and DPO.

| Training Method | Train Samples/s | Train Steps/s | Total FLOS | Run Time/s |
|---|---|---|---|---|
| SFT | 1.035 | 0.129 | $2.82 \times 10^{17}$ | 8848 |
| DPO | 0.979 | 0.123 | $6.22 \times 10^{17}$ | 9346 |
| $\Delta$ | -0.056 | -0.006 | $+3.4 \times 10^{17}$ | +498 |

As can be seen from the table, in actual computation, the computational cost between SFT and DPO is not significantly different. For the computation of 9,000 pieces of data, the total computation time only increased by approximately 8.3 minutes.

---

[6]https://github.com/ModalMinds/MM-EUREKA
[7]https://github.com/hiyouga/LLaMA-Factory

# G   QUALITATIVE EXAMPLES

---

### QA Case Analysis: Case #001

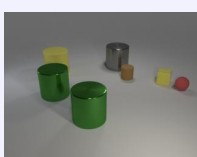

**Question:** Hint: Please answer the question requiring an integer answer and provide the final value, e.g., 1, 2, 3, at the end. Question: Subtract all yellow matte blocks. Subtract all tiny brown cylinders. How many objects are left?

**Ground Truth:** 5

**Qwen2.5-VL-7B response:** 6

- - - - - - - - - - - - - - - - - - - - - - - - - - - - - - - - - - - - - - - - - - - - -

#### Ours-7B response

<think>To solve the problem, first identify all objects in the image:
1. Yellow matte cylinder
2. Green metallic cylinder (left)
3. Green metallic cylinder (front)
4. Silver metallic cylinder
5. Tiny brown cylinder
6. Yellow matte block
7. Red matte sphere
Next, follow the instructions:
1. Subtract all yellow matte blocks: Removing the yellow matte block leaves 6 objects.
2. Subtract all tiny brown cylinders: Removing the tiny brown cylinder leaves 5 objects.
Thus, the remaining objects are 5.
</think>
<answer>The answer is 5 </answer>

#### Qwen2.5-VL-7B response

To solve this, we first identify all objects in the image: there are 8 objects in total. The yellow matte block is one object, and the tiny brown cylinder is another. Subtracting these two objects leaves us with $8 - 1 - 1 = 6$ objects.
So, the number of objects left is 6.

### QA Case Analysis: Case #002

**Web pages visited**

| Person | Number of Web pages |
|--------|---------------------|
| Bonnie | 84 |
| Winston | 56 |
| Percy | 70 |
| Derek | 68 |
| Jake | 60 |
| Finn | 47 |
| Carter | 75 |
| Kurt | 52 |

**Question:** Hint: Please answer the question requiring an integer answer and provide the final value, e.g., 1, 2, 3, at the end. Question: Several people compared how many Web pages they had visited. What is the mean of the numbers?'

**Ground Truth:** 64

**Qwen2.5-VL-7B response:** 67.75

---

#### Ours-7B response

<think>To find the mean of the numbers, first sum all the numbers and then divide by the count of numbers.
The numbers are: 84, 56, 70, 68, 60, 47, 75, 52.
Sum of the numbers:

$$84 + 56 + 70 + 68 + 60 + 47 + 75 + 52$$

Calculating step by step:

$$84 + 56 = 140$$
$$140 + 70 = 210$$
$$210 + 68 = 278$$
$$278 + 60 = 338$$
$$338 + 47 = 385$$
$$385 + 75 = 460$$
$$460 + 52 = 512$$

The sum of the numbers is 512.
Next, count the number of values: There are 8 values.
The mean is the sum divided by the count:

$$\text{Mean} = \frac{512}{8} = 64$$

So, the mean of the numbers is 64.
</think>
<answer>The answer is $\boxed{64}$</answer>

#### Qwen2.5-VL-7B response

To find the mean of the numbers, we need to follow these steps:
1. Add up all the numbers in the list. 2. Count how many numbers there are. 3. Divide the sum by the count.
Let's start with step 1: Add up all the numbers.
The numbers are: 84, 56, 70, 68, 60, 47, 75, 52.
**Sum = 84 + 56 + 70 + 68 + 60 + 47 + 75 + 52**
**Sum = 542**
...
Mean = 542 / 8
Mean = 67.75
So, the mean of the numbers is 67.75.

## QA Case Analysis: Case #003

**Question:** Hint: Please answer the question and provide the correct option letter, e.g., A, B, C, D, at the end. Question: At a certain moment, there is a passenger ship at sea point P, and lighthouse A is measured in the direction 30.0 north by east of P, and is 50.0 nautical miles away. The passenger ship sails at the speed of 60.0 nautical mile/hour in the direction of 60.0 from north by west for $\frac{2}{3}$ hours to reach point B, then tan∠BAP =

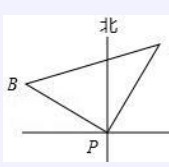

Choices:
(A) $\frac{4}{5}$ (B) $\frac{6}{5}$ (C) $\frac{\sqrt{5}}{5}$ (D) $\frac{2\sqrt{5}}{5}$

**Ground Truth:** (A) $\frac{4}{5}$

**Qwen2.5-VL-7B response:** (B) $\frac{6}{5}$

---

### Ours-7B response

<think>First, we need to determine the coordinates or distances relevant to the problem. The initial position of the passenger ship is at point $P$, and it is 50 nautical miles away from lighthouse $A$ in the direction 30.0 degrees north by east. The passenger ship then sails in the direction 60.0 degrees from north by west for $\frac{2}{3}$ hours at a speed of 60 nautical miles per hour, reaching point $B$.

The distance $PB$ can be calculated as follows:

$$PB = 60 \times \frac{2}{3} = 40 \text{ nautical miles}$$

The direction 60.0 degrees from north by west means that the angle $\angle BPA = 30.0° + 60.0° = 90.0°$. Therefore, $\triangle ABP$ is a right triangle with $\angle BAP = 30.0°$. In a right triangle, the tangent of an angle is the ratio of the opposite side to the adjacent side. Here, $\tan\angle BAP = \frac{BP}{AP}$.

Since $AP = 50$ nautical miles and $PB = 40$ nautical miles, we can use the Pythagorean theorem to verify:

$$AB = \sqrt{AP^2 + BP^2} = \sqrt{50^2 + 40^2} = \sqrt{2500 + 1600} = \sqrt{4100} = 10\sqrt{41}$$

However, we only need the ratio $\frac{BP}{AP}$:

$$\tan\angle BAP = \frac{BP}{AP} = \frac{40}{50} = \frac{4}{5}$$

Thus, the answer is $\boxed{A}$.</think>

<answer>The answer is $\boxed{A}$</answer>

---

### Qwen2.5-VL-7B response

To solve this problem, we need to determine the coordinates of points A and B relative to point P, and then find the tangent of the angle ∠BAP.
1. Determine the coordinates of point A relative to point P:
...
2. Determine the coordinates of point B relative to point P:
...
3. Calculate the tangent of ∠BAP:
... - Simplifying the expression, we get $\frac{625\sqrt{3}-500-1250\sqrt{3}-1500}{625\sqrt{3}+1500-625\sqrt{3}+500} = \frac{-925\sqrt{3}-1000}{2000} = \frac{-925\sqrt{3}-1000}{2000} = \frac{-925\sqrt{3}-1000}{2000} = \frac{-925\sqrt{3}-1000}{2000} = \frac{6}{5}$.
Therefore, the correct answer is $\boxed{B}$.

