# OpenReview forum: "SPECS: Decoupling Multimodal Learning via  Self-distilled Preference-based Cold Start"
_ICLR.cc/2026/Conference — ICLR 2026 Poster_

### Official Review · Reviewer_8EWE · 2025-10-19

**Soundness:** 3
**Presentation:** 3
**Contribution:** 3
**Rating:** 8
**Confidence:** 3

**Summary:**

Currently, with the fast development of RL, its application has expanded to MLLM. Most representative paradigms begin with a cold start under supervised fine-tuning (SFT). A challenge here is SFT cold start may induce instruction-style overfitting, weaken out-of-distribution generalization, and ultimately affect downstream RL. In this view, the authors first introduce the Generalization Factor (GF) coefficient to quantify the generalization capability under different methods, showing that preference-based training can do better than SFT. This further brings a Self-distilled, Preference-based Cold Start framework (SPECS) in preference-based training style.

**Strengths:**

1. The motivation of this paper is sound, and the observation on SFT cold start is fundamental. The GF criteria, though without significant novelty, is straightforward and useful.

2. After raising the problem, the authors propose an intuitive and effective approach to solving the problem.

3. The paper is easy to follow.

**Weaknesses:**

1. The self-distilled dataset needs further discussion. In Sec. 3.2, the rejected responses are responses that also contain the correct answer, but deviate from the required format. In stage 1 training, is format deviation the other problem we need to consider? What about wrong answers, wrong thinking process, etc? From my perspective, the current design might face the challenge of hard negative mining. More discussions are needed.

**Questions:**

The paper has the finding that stage 1 rl can benefit following rl training. The experiments are sound and the discussions are solid. Overall, it is a good paper.

---

> ### Author Response · Authors · 2025-11-18
> **Reply**
>
> Thanks for the thoughtful comment. One of the important reasons why we use preference data is that preference data can better focus on learning objectives through positive and negative data, while the supervised data of SFT mixes various learning objectives during training. During the cold-start phase, we focus on enabling the model to learn formatted responses and certain reasoning paradigms. In this way, a relatively simple and single learning objective can create a more suitable cold-start model for the subsequent RL phase. Our ideas have also been verified through experiments. In the ablation study, we obtained coupled data by mixing rejected responses that have correct answers but incorrect formats and those that have incorrect answers and incorrect formats. The results trained with the same training parameters are as follows, which are from Table 3 in Section 4.3 of the original paper.
>
> | Model               | MegaBench | MMMU      | MathVista | MathVision | MathVerse | Eval          |
> | ------------------- | ------------- | ------------- | ------------- | -------------- | ------------- | ------------- |
> | Qwen2.5-7b-instruct | 35.07         | 54.2          | 63.70         | 25.40          | 38.20         | 43.31         |
> | Coupled Data        | 37.02 / 38.76 | 55.44 / 55.44 | 71.10 / 73.10 | 27.37 / 28.65  | 47.46 / 47.46 | 47.67 / 48.68 |
> | Decoupled Data      | 37.52 / 39.17 | 54.89 / 56.78 | 72.00 / 75.90 | 25.75 / 29.50  | 46.19 / 48.73 | 47.27 / 50.02 |
>
> It can be seen that after cold start (the previous value), the model trained with coupled data shows a greater improvement, which is related to its mixed learning objectives during the cold start phase. However, after GRPO, the Decoupled data model surpasses the coupled data model instead. This is an interesting finding, which precisely indicates that a single objective in the cold start phase is more suitable as the starting point of training for subsequent RL.
>
> The hard negative mining you mentioned is also a very good point. It is a very important issue, but in our practice, we adopt the scheme of correct answer data collection + format pollution (see Section 3.2 in the original paper) to maximize the data utilization of obtaining rejected responses, and this process is not difficult during the experiment.
>
> Thank you again for your valuable suggestions.

---

### Official Review · Reviewer_hLrc · 2025-10-27

**Soundness:** 2
**Presentation:** 3
**Contribution:** 2
**Rating:** 4
**Confidence:** 4

**Summary:**

This paper proposes SPECS, a framework for training multimodal models using DPO and RL. The framework involves first generating preference data from the model, training it on the preference data with DPO, and then performing GRPO training. The authors show that using DPO is preferable to SFT for training an initial RL checkpoint due to its superior generalization, and yields more stable RL training. Their final model outperforms solid baselines or training using only RL.

**Strengths:**

- Model outperforms prior work and the additional DPO is clearly helpful.
- Ablations show the distillation strategy is quite effective
- Showing DPO generalizes better than SFT with the generalization gap metric is interesting.

**Weaknesses:**

- I think the work needs more careful ablations, particularly around the self-distillation.
  - Gemini-flash is used as the LM judge when choosing responses. How does this compare to just distilling from Gemini? Is using it as a judge cheaper?
  - The authors do some additional RL to create a model to distil from and claim “[it] is more adept at exploring the solution space”. What is the justification for this? Does distilling directly from the base model (with some careful prompting) perform worse? A stronger justification here would be nice.
  - In table 3, you claim that decoupled outperforms coupled data for cold-starting, but it appears it underperforms the coupled data (slightly) pre-RL training, but does indeed outperform post-RL training. It would be useful to explain why this is! Similarly, considering that the average difference is only around 1.5 points, it would be useful to know what the average noise on the dataset is (i.e., some sort of statistical significance measure on the results - is 1.5 within noise or not?).
- The generalization gap metric is interesting, but its connection with better cold start checkpoints seems tenuous. If we do RL training on only in-domain prompts, does it matter that one method has better OOD performance? A more concrete comparison where you train an SFT model with RL using the same setup as the DPO+GRPO model would be more convincing. The authors have clearly trained an SFT+RL model (used for analysis in Section 4.5), so hopefully it would just be a matter of evaluating that.
- There is some interesting analysis in Appendix A.4 that suggests the DPO better improves pass @ k, but as far as I can see this is never mentioned in the main text (not even a reference to that appendix section itself!)
- The authors note that the DPO-trained model is more stable during RL training, but don’t provide an explanation as to why. It’d be useful to see some sort of analysis or justification for this - perhaps it suggests the hyperparameters for RL training on the SFT model are suboptimal?
- The authors do not give enough details on their experimentation. How many training steps / epochs was training run for for DPO/SFT/RL? For RL, what KL penalty was used, and how long were responses allowed to be? It would also be useful to get some indication of how long SFT/DPO/RL training took in GPU-hours.

Overall, I think the work is reasonable, and the finding around using DPO instead of SFT for cold start is interesting. But more justification around decisions are required, and some clearer ablations.

**Questions:**

See weaknesses.

---

> ### Author Response · Authors · 2025-11-18
> **Reply (Part 1 / 2)**
>
> Thank you very much for your valuable comments and suggestions. Below we  address each of them in detail.
>
> **W1: The supplement of the ablations**
>
> **W1.1: Regarding data filtering**
>
> Yes, this is a good and important point. We think it is crucial to ensure the quality and accuracy of chosen response in cold start parse. However, we found that there are a small number of cases (generated by GRPO-zero) in chosen response where the answers are not consistent with the reasoning, which could affect training, such as the following case:
>
> > \<think\>To determine the total number of dogs in the picture, we need to count each dog individually. From the image, we can see that there are **4** dogs. \</think\> \n \<answer\> The answer is \boxed{**3**} \</answer\>
>
> The reasons for this situation still need to be explored, and we believe it may be related to reward hacking in GRPO training (because the answer to this training data was incorrectly labeled as 3). To prevent these data from affecting the cold start training, we use another LLM to verify the data. This evaluation process is simple and objective, with no significant errors, and will not introduce bias into data filtering. The table below shows the differences in the final model performance between the data filtered by the advanced LLM and the randomly filtered data:
>
> | Model                        | MegaBench| MMMU | MathVista | MathVision | MathVerse | Eval  |
> | ---------------------------- | --------------- | ------------ | -------------- | ---------- | -------------- | ----- |
> | Qwen2.5-7b-instruct          | 35.07           | 54.2         | 63.70          | 25.40      | 38.20          | 43.31 |
> | w.o verify | 38.72           | 55.11        | 73.5           | 26.90      | 46.44          | 48.13 |
> | w. verify  | 39.17           | 56.78        | 75.90          | 29.50      | 48.73          | 50.02 |
>
> We do not choose to directly use Gemini for data distillation. On the one hand, this approach is far more costly than using Gemini for verification. On the other hand, it is because, as we proved in Section 4.3 of the original paper, **“Preference data closely aligned with the model’s intrinsic capability distribution is more effective for alignment than guidance from a more capable but dissimilar external model.”** We also conducted experiments using Gemini to distill data with the same training method, and the results are as follows：
>
> | Model               | MegaBench | MMMU  | MathVista | MathVision    | MathVerse | Eval          |
> | ------------------- | --------------- | ------------- | -------------- | ------------- | -------------- | ------------- |
> | Qwen2.5-7b-instruct | 35.07           | 54.2          | 63.70          | 25.40         | 38.20          | 43.31         |
> | Gemini Distillation | 32.40 / 37.46   | 52 / 52.67    | 65.7 / 74.9    | 21.34 / 26.05 | 34.51 / 42.63  | 41.19 / 46.74              |
> | Self Distillation   | 37.52 / 39.17   | 54.89 / 56.78 | 72.00 / 75.90  | 25.75 / 29.50 | 46.19 / 48.73  | 47.27 / 50.02 |
>
> **W1.2 Introduction to the GRPO-zero model**
>
> Thanks for a thoughtful suggestion. The insight of using RL directly on the model to improve its reasoning ability comes from DeepSeek-r1. At the same time, we have conducted statistics on some indicators of the responses of the original model and the zero model to the training data questions, and the statistical results are as follows:：
>
> | Model               | Format acc/% | Answer acc/% | Num of reasoning words / 1k chars |
> | ------------------- | ------------ | ------------ | -------------------------------------- |
> | Qwen2.5-7b-instruct | 41.62        | 30.42%       | 4.26                                   |
> | Our-GRPO-zero       | 96.74        | 52.82%       | 4.99                                   |
>
> Obviously, GRPO-zero has higher data utilization in the collection of chosen responses due to its higher format accuracy and answer accuracy. At the same time, we counted the number of reasoning words (including transition words, causal words, sequential words, etc.) per 1k characters for both models. We also used the same data collection method to collect responses with correct formatting and correct answers distilled from the base model as chosen responses (this is much more difficult than distilling from GRPO-zero), and the results are as follows:
>
> | Model                     | MegaBench     | MMMU          | MathVista     | MathVision    | MathVerse     | Eval          |
> | ------------------------- | ------------- | ------------- | ------------- | ------------- | ------------- | ------------- |
> | Qwen2.5-7b-instruct       | 35.07         | 54.2          | 63.70         | 25.40         | 38.20         | 43.31         |
> | distillation by base      | 35.37 / 37.92 | 53.11 / 56.11 | 67.9 / 74.4   | 25.55 / 28.68 | 43.40 / 46.82 | 45.07 / 48.79 |
> | distillation by grpo-zero | 37.52 / 39.17 | 54.89 / 56.78 | 72.00 / 75.90 | 25.75 / 29.50 | 46.19 / 48.73 | 47.27 / 50.02 |

---

> ### Author Response · Authors · 2025-11-18
> **Reply (Part 2 / 2)**
>
> **W1.3 Regarding coupled data**
>
> Thank you for identifying this. We believe that the performance of a model after rl is not necessarily positively correlated with its performance after cold start. From the experimental results, an interesting phenomenon emerges: the use of Coupled Data enables the model to achieve better performance during the cold start phase, yet after GRPO, its performance is weaker than that of the model cold-started with Decoupled Data. We attribute this to the fact that Coupled Data links two learning objectives—reasoning paradigms and answer accuracy—whereas Decoupled Data focuses solely on reasoning paradigms during the cold start phase. This also indicates that focusing solely on a single learning objective during the cold start phase provides greater potential for the RL phase.
>
> Regarding the noise in the evaluation data, we believe that the evaluation results are credible. On the one hand, we used greedy generation with a temperature of 0 during evaluation to avoid evaluation randomness. On the other hand, in datasets with an average size of over 1,000, for models of size 7B, an average improvement of 1.5 across various benchmarks is significant. Finally, we also analyzed some benchmarks such as MMMU. Although the overall accuracy difference is not significant (55.33% vs 56.86%), a stratified analysis shows that our model has a significant improvement in the Science category (+7.43%, p=0.015, n=175). This improvement was averaged out, indicating that the actual improvement of our method on the model in scientific reasoning tasks is not caused by evaluation noise.
>
> **W2: Comparative experiments between SFT+RL and DPO+RL**
>
> We present our findings in Section 4.4 of the original paper. By comparing various cold-start methods, we found that the GF before RL are somewhat positively correlated with the final performance of the model to a certain extent. I believe that in most scenarios, the goal of model training is not just to improve the model's ID Performance. How to slow down the model's forgetting of OOD tasks during training is also an object of research for many scholars. Inspired by your reply, we conducted a comparative experiment between SFT + RL and DPO + RL. Please note that the SFT data here is the chosen from the DPO data (derived from the self-distillation in our work), rather than the open-source or other model-distilled data used in most SFT works. The experimental results are consistent with our expectations.
>
> | Model               | MegaBench | MMMU  | MathVista | MathVision | MathVerse | Eval  |
> | ------------------- | --------- | ----- | --------- | ---------- | --------- | ----- |
> | Qwen2.5-7b-instruct | 35.07     | 54.2  | 63.70     | 25.40      | 38.20     | 43.31 |
> | SFT-GRPO            | 37.52     | 54.44 | 74.10     | 28.61      | 43.60     | 47.65 |
> | DPO-GRPO            | 39.17     | 56.78 | 75.90     | 29.50      | 48.73     | 50.02 |
>
> **W3：Regarding Appendix A.4.**
>
> Thanks. There is an incorrect reference to a serial number here. We will correctly cite and elaborate on the content of this section in the updated main text of the paper.
>
> **W4:  Regarding the analysis of training stability.**
>
>  We ensure that all training parameters and training frameworks for SFT training and DPO training are consistent, and we believe this can maximize the fairness of the comparison. By observing the training logs, we found that compared with DPO cold start, the policy loss of SFT cold start fluctuates more, and the overlong penalty and format reward of the responses generated during its training process are also unstable. In further analysis of possible reasons, we believe this is related to the training objectives of the cold-start phase and the RL phase. The SFT training objective is to maximize log likelihood, which is a form of imitation learning, while the learning objective of DPO can be seen as directly optimizing an implicit reward model consistent with preference data, which is more aligned with the subsequent reward-driven GRPO optimization objective. Therefore, using a DPO-based model as a starting point also brings about more stable training for subsequent RL.
>
> **W5：Supplementary details of the experiment.**
>
> We presented some details of the training experiments in Section 4.1 of the original paper, but as you correctly pointed out, this is not entirely complete. In the appendix F of the revised paper, we have provided more training details, including those you mentioned. Additionally, to ensure the double-blind principle in the rebuttal, we have not attached the open-source code link. After the rebuttal ends, we will upload all training data and training scripts to Github and Huggingface.
>
> Thank you again for your valuable suggestions. We will refine the main text and appendix of the revised paper accordingly.

---

> ### Comment · Reviewer_hLrc · 2025-11-21
>
> Thank you for your response!
>
> W1.1 -> The ablation with the distillation data from gemini makes the point much clearer, thank you! I think this is an important comparison.
>
> W1.2 -> So it seems not that they model is necessarily better at "exploring" the solution space, but rather just better at adhering to the output formats? Does the GRPO-zero model have higher answer accuracy because it adheres to the format more? I think the motivation for using GRPO-zero in the paper is still unclear. I guess you are doing something like: we need to distil from a model 'close' to our current base. By training the base a little with GRPO, we end up with a model that is still 'close' but has higher performance and so is easier to distil from, is that right? Spelling out this motivation is useful, since its not necessarily clear that the motivation from deepseek applies in your setting (and in the deepseek paper, the cold-start data is mainly presented as a way to collect lots of high-quality data cheaply).
>
> W1.3 -> thanks, this statistical analysis is useful.
>
> W4 -> I guess my point here was that, it may be that if you sweep hyperparameters more thoroughly you can find a better setup for SFT. How were your current hyperparameters chosen? If you swept them only for your method, or tuned them over experimentation, it may present an unfair bias to SFT even though you keep the hyperparameters consistent between the two methods - you may be accidentally using some hyperparameters just better suited to your approach. For example, I've found that some runs unstable with a clip-higher of 0.28 are NOT unstable when you lower it to 0.27, or 0.272. See https://arxiv.org/abs/2510.13786 A.17.1 for more experiments around the general instability of clip-higher. But I also understand this can be expensive, so just some further justification of why you chose e.g. 400 steps of training and not longer, why you chose a LR of 1e-6, would be useful.
>
> You also claim "while the learning objective of DPO can be seen as directly optimizing an implicit reward model consistent with preference data, which is more aligned with the subsequent reward-driven GRPO optimization objective", could you elaborate on this a bit more?
>
> Thank you a lot for the extra details and the promise to update the paper with further experimental details! Since that was my main issue, I've raised my score a bit, although I am still confused by some of the points in your rebuttal as explained above.

---

> ### Author Response · Authors · 2025-11-22
> **Response to Follow-up Questions**
>
> Thank you very much for your positive feedback!
>
> **W1.2 Motivation of GRPO-zero**
>
> Your understanding of our method is very insightful and accurate. The main reason we chose GRPO-zero for data generation is that it is closer to the base model and achieves better performance. Specifically, compared with SFT or other base models, GRPO-zero—when directly trained via RL—exhibits a smaller shift in the token probability distribution [1]. Thus, distilling from GRPO-zero also aligns with the concept of self-distillation.
>
> This superior performance is reflected in two key aspects. On the one hand, it is reflected in the data utilization rate we discussed in Reply 1. On the other hand, it helps improve the quality of the chosen responses. We refer to this improvement in response quality as an enhanced ability to "explore" the solution space, which includes not only its stronger adherence to formatting requirements but also its improved reasoning capabilities after RL training (This can be verified by higher answer accuracy, an increased number of reasoning-related words, and findings from existing works [2-4]).
>
> We acknowledge your observation that our explanation of this motivation was insufficiently clear. We will provide a more explicit explanation of the motivation for using GRPO-zero in the revised manuscript.
>
> [1] Fu Y, Chen T, Chai J, et al. SRFT: A Single-Stage Method with Supervised and Reinforcement Fine-Tuning for Reasoning[J]. arXiv preprint arXiv:2506.19767, 2025.
>
> [2] Peng Y, Zhang G, Zhang M, et al. Lmm-r1: Empowering 3b lmms with strong reasoning abilities through two-stage rule-based rl[J]. arXiv preprint arXiv:2503.07536, 2025.
>
> [3] Zhang J, Huang J, Yao H, et al. R1-vl: Learning to reason with multimodal large language models via step-wise group relative policy optimization[J]. arXiv preprint arXiv:2503.12937, 2025.
>
> [4] Zeng W, Huang Y, Liu Q, et al. Simplerl-zoo: Investigating and taming zero reinforcement learning for open base models in the wild[J]. arXiv preprint arXiv:2503.18892, 2025.
>
> **W4 Regarding the selection of parameters**
>
> Thank you for pointing out such valuable information for reference.
>
> - The setting of the number of steps is derived from our observations on the current training set. The factors influencing our selection are as follows: 1) The model is fully trained until convergence. 2) The training data volume is the same across various methods. 3) On the basis of convergence, the minimum number of training steps is selected.  As can be seen in Figure 4 of our original paper (other benchmarks show similar patterns), we did not choose the training step count at the moment when the gap between the two methods was the largest (800 steps). Instead, after comprehensively considering the above points, we chose to train for 400 steps. Additionally, in preliminary experiments, we observed that when the number of training steps exceeds 400, the model begins to produce repetitive outputs. Therefore, choosing this number of steps is the optimal balanced choice.
>
> - Regarding other parameters such as the learning rate, we referred to the parameters used in most excellent works, such as [5, 6] (also for the sake of fair evaluation), and did not perform additional tuning for any training algorithm. However, your feedback and sharing have given us important insights into more rigorous training experiment.
>
> - Regarding the consistency of training objectives. An interesting point is that both DPO and GRPO aim for contrastive loss, and they both optimize strategies by comparing positive and negative samples [7]. Their main difference lies in how these positive and negative samples are defined and weighted. DPO uses paired preference data, while GRPO implicitly defines positive and negative samples within a group through reward normalization. In other words, DPO can be approximated as GRPO with n = 2. These two are more consistent in their training objectives compared to SFT and GRPO. And we believe that one of the reasons for this training stability is the consistency of objectives in multi-stage training.
>
> [5] Meng F, Du L, Liu Z, et al. Mm-eureka: Exploring the frontiers of multimodal reasoning with rule-based reinforcement learning[J]. arXiv preprint arXiv:2503.07365, 2025.
>
> [6] Ma Y, Du L, Shen X, et al. One RL to See Them All: Visual Triple Unified Reinforcement Learning[J]. arXiv preprint arXiv:2505.18129, 2025.
>
> [7] Wu Y, Ma L, Ding L, et al. It Takes Two: Your GRPO Is Secretly DPO[J]. arXiv preprint arXiv:2510.00977, 2025.
>
>
> Thank you for your professional suggestions. We will add a clearer explanation of the motivation behind GRPO-zero and the basis for hyperparameter selection to the revised paper.

---

> > ### Comment · Reviewer_hLrc · 2025-11-27
> >
> > Thanks for the additional reply and adding such details into the paper! I'm keeping my positive score :)

---

### Official Review · Reviewer_bc3X · 2025-10-30

**Soundness:** 2
**Presentation:** 2
**Contribution:** 2
**Rating:** 2
**Confidence:** 4

**Summary:**

This paper proposes SPECS, a cold start strategy for preference-based multimodal learning. They first introduce a generalization factor metric to evaluate models' generation capacity. They then design a 3-stage learning framework: 1) generate format-aware preference data via self-distillation; 2) DPO cold start for learning the format consistency; 3) GRPO to improve deep reasoning.

**Strengths:**

- Experiments demonstrate the performance gain of the proposed method against selected base models and vanilla GRPO on various benchmarks.
- The paper is overall clearly written and easy to follow.
- Ablation study is conducted to validate the performance of decoupled (format) DPO data.

**Weaknesses:**

- The paper is not technically sound. It largely reaffirms the well-established observation that RL-based methods improve generalization compared to supervised finetuning,  but does not offer significant new technical insight.
- The introduction of the zero model (trained with GRPO before self-distillation) appears unnecessary. It is unclear why the chosen and rejected samples cannot be directly generated from the base model, especially since both are later used for format-level preference training.
- The definition of Generalization Factor and lack clarity theoretical justification. The formula in L131–133 appears confusing and unsimplified, possibly due to a typo.
- The proposed method is costly. The cold start stage requires GRPO training on 86k data, as well as gemini-2.5-flash as strong LLM-as-a-judge.
- The effectiveness of DPO cold start is not directly validated; it would be helpful to compare with a control baseline that performs SFT cold start + RL using the same chosen data to isolate the benefit of preference-based training.

**Questions:**

- For the rejected samples, why are rollouts taken from the base model instead of the exploratory GRPO-zero model?
- Could the authors clarify the mathematical definition and interpretation of GF?
- Why is the zero-model necessary for self-distillation? Given the strong LLM-as-a-Judge, can we directly sample from base model rollout?
- Since you use both SFT and DPO loss for cold start training, how is the format decoupled?

---

> ### Author Response · Authors · 2025-11-18
> **Reply (Part 1 / 3)**
>
> Thank you very much for your valuable comments and suggestions. Below we  address each of them in detail.
>
> **W1: Regarding the contributions of the paper.**
>
> The reviewer regards the core contributions of our work as:  "an observation that RL-based methods improve generalization compared to supervised finetuning".  However, our paper focuses more on the training cold start of VLMs to improve the model’s reasoning ability. We propose that in cold start, self-distilled decoupled preference data focusing on a single learning objective have advantages over most current works in terms of the model training of subsequent RL for the first time.
>
> In addition to the analysis of the generalization of DPO and SFT (Section 2), more space is devoted to putting forward some new technical insights: a method for constructing self-distilled decoupled preference data (Section 3.2), a DPO-based cold start scheme (Section 3.3), the significance of data decoupling and self-distillation for training (Section 4.3), the impact of model generalization on model GRPO (Section 4.5), and some interesting findings are also elaborated in the appendix, including the analysis of mixed loss of DPO and SFT (Section A.3.2 of the original paper), and the analysis of the potential of models trained by DPO and SFT (Section A.4 of the original paper). I believe these findings and technical insights can provide more thoughts and inspirations for the industry regarding model training. However, we also deeply recognize some omissions in the analysis and experiments, and we will update and improve them in the latest version of the paper. We look forward to your attention.
>
> **W2 & Q3: Introduction to the GRPO-zero model.**
>
> Regarding your question, we indeed lack more analysis in the paper, but I would like to emphasize that the introduction of GRPO-zero is necessary both intuitively and in specific experiments. As we elaborated in Section 1 of the paper, the insight for introducing GRPO-zero comes from DeepSeek-r1-zero, a highly successful approach that improves model reasoning capabilities through reinforcement learning. We aim to improve the data utilization rate and data quality obtained from chosen responses by introducing GRPO-zero. For comparison, We have conducted statistics on some indicators of the responses of the original model and the zero model to the training data questions, and the statistical results are as follows:
>
> | Model               | Format acc/% | Answer acc/% | Number of reasoning words / 1k chars |
> | ------------------- | ------------ | ------------ | -------------------------------------- |
> | Qwen2.5-7b-instruct | 41.62        | 30.42%       | 4.26                                   |
> | GRPO-Zero      | 96.74        | 52.82%       | 4.99                                   |
>
> Obviously, GRPO-Zero has higher data utilization in the collection of chosen responses due to its higher format accuracy and answer accuracy. At the same time, we counted the number of reasoning words (including transition words, causal words, sequential words, etc.) per 1000 characters for both models. Our ours-grpo-zero model’s responses match the reasoning paradigm we want much better, and they have more reasoning words, and therefore, it is more suitable as a source for chosen responses. We also used the same data collection method to collect responses with correct formatting and correct answers distilled from the base model as chosen responses (this is much more difficult than distilling from ours-grpo-zero), and the results are as follows:
>
> | model                     | MegaBench     | MMMU          | MathVista     | MathVision    | MathVerse     | Eval          |
> | ------------------------- | ------------- | ------------- | ------------- | ------------- | ------------- | ------------- |
> | Qwen2.5-7b-instruct       | 35.07         | 54.2          | 63.70         | 25.40         | 38.20         | 43.31         |
> | distillation by base      | 35.37 / 37.92 | 53.11 / 56.11 | 67.9 / 74.4   | 25.55 / 28.68 | 43.40 / 46.82 | 45.07 / 48.79 |
> | distillation by grpo-zero | 37.52 / 39.17 | 54.89 / 56.78 | 72.00 / 75.90 | 25.75 / 29.50 | 46.19 / 48.73 | 47.27 / 50.02 |
>
> It can be seen from the above experimental results that if the base model is directly used for distillation, this process is not only more difficult (due to inefficient data utilization), but the final results also decrease due to the quality of the data.

---

> ### Author Response · Authors · 2025-11-18
> **Reply (Part 2 / 3)**
>
> **W3 & Q2: Definition of GF.**
>
> Thank you for your valuable suggestions. Our definition of GF is simple and theoretically sound. To put it in one sentence, we define GF as the $F_\beta$-score  of the model with respect to OOD performance gains and ID performance gains. The reason for adopting this metric is that the $F_\beta$-score is particularly suitable for average ratios. Its most prominent feature is that the result tends to lean toward the smaller number. This perfectly aligns with our needs: as long as either the ID or OOD performance is very poor, the final score will be very low. We can also control the size of  to reflect the degree of importance we attach to OOD performance gains during the training process. This is not subjectively defined, but is mathematically meaningful and interpretable. However, it is true that our expression in the paper is unclear, and there are some issues with individual formulas. In the revised paper, we will better clarify tion of GF.
>
> **W4: Regarding method cost.**
>
> In fact,  compared with  other equally excellent works, our method requires much less in terms of both data volume and other expenses:
>
> - Regarding the amount of cold-start data. Please note that the data we used in the cold-start training of GRPO-zero and the final reinforcement learning process in the third stage is the same, which comes from Orsta47K and virl39K, and the cold-start data is sampled from these data. That is to say, on the basis of surpassing so many baselines, the only data we used is this 86K.
>
> - Regarding the use of gemini-2.5-flash. The main purpose of using Gemini-2.5 flash for data filtering is to screen out data where the answers are consistent with the reasoning, avoiding situations where the grpo-zero reasoning process is inconsistent with the answers, which could affect training. This evaluation process is simple ,objective and cheap. In terms of specific costs, we have conducted rigorous statistics. The average number of input tokens is 2991.4 (using the Qwen2.5VL-7B-Instruct tokenizer), and a total of 12k data were filtered. The total number of requests is 35,896,800 tokens. Excluding the free tier plan of the Gemini-2.5-flash API, the paid tier plan is 0.3 dollars per million tokens, so the total input cost is approximately 10.7 dollars. We only required it to reply with 1 or 0. The total usage cost does not exceed 15 dollars. In fact, due to the simplicity of the task, we can completely consider using cheaper models or completely free open-source models for screening and filtering tasks.
>
> - In other similar works, Vision-R1 [1] constructed 200k long COT cold-start data from 360K data through DeepSeek-r1 during cold start, and additionally constructed 10K RL training data from multiple open-source datasets. R1-Onevision [2] generated 155K long COT data from numerous datasets using GPT-4o, DeepSeek-r1, EasyOCR, and others.
>
> **W5: Comparative experiments between SFT+RL and DPO+RL**
>
> We present our findings in Section 4.4. By comparing various cold-start methods, we found that the GF before RL are somewhat positively correlated with the final performance of the model to a certain extent. I believe that in most scenarios, the goal of model training is not just to improve the model's ID Performance. How to slow down the model's forgetting of OOD tasks during training is also an object of research for many scholars. Inspired by your reply, we conducted a comparative experiment between SFT + RL and DPO + RL. Please note that the SFT data here is the chosen from the DPO data (derived from the self-distillation in our work), rather than the open-source or other model-distilled data used in most SFT works. The experimental results are consistent with our expectations.
>
> | Model               | MegaBench | MMMU  | MathVista | MathVision | MathVerse | Eval  |
> | ------------------- | --------- | ----- | --------- | ---------- | --------- | ----- |
> | Qwen2.5-7b-instruct | 35.07     | 54.2  | 63.70     | 25.40      | 38.20     | 43.31 |
> | SFT-GRPO            | 37.52     | 54.44 | 74.10     | 28.61      | 43.60     | 47.65 |
> | DPO-GRPO            | 39.17     | 56.78 | 75.90     | 29.50      | 48.73     | 50.02 |
>
> [1] Huang W, Jia B, Zhai Z, et al. Vision-r1: Incentivizing reasoning capability in multimodal large language models[J]. arXiv preprint arXiv:2503.06749, 2025.
>
> [2] Yang Y, He X, Pan H, et al. R1-onevision: Advancing generalized multimodal reasoning through cross-modal formalization[J]. arXiv preprint arXiv:2503.10615, 2025.

---

> ### Author Response · Authors · 2025-11-18
> **Reply （Part 3 / 3）**
>
> **Q1: Regarding preference data generation**
>
> Similar to the analysis in **W2**, considering data utilization and the differences in response formats and reasoning paradigms between the GRPO-zero model and the base model itself, it is intuitive for us to select the base model to generate rejected responses and the GRPO-zero model to generate chosen responses.
>
> **Q4: Regarding format decoupling**
>
> We added the integration of SFT loss on top of DPO to enhance the regularization effect on the chosen response. More analysis is provided in Appendix A.3.2 of the original paper. We believe this does not affect the decoupling of formats in the construction of preference data (Section 3.2), as it can be observed that during the training process, the reward margins for the decoupled data focused on format objectives are continuously increasing, which is completely different from the coupled training that solely uses SFT loss.
>
> Thank you again for your valuable comments. We will refine the main text and appendix of the revised paper accordingly.

---

> ### Author Response · Authors · 2025-11-26
> **Gentle Reminder: Feedback on Response**
>
> Dear reviewer bc3X：
>
> We are truly grateful for your constructive comments on our submission.
>
> In response to your review, we have added extensive experiments to clarify and answer your concerns. We conducted a more in-depth analysis of the use of GRPO-zero in our method and carried out corresponding ablation experiments to illustrate the importance of the GRPO-zero model in generating preference data. We also performed a more comprehensive experimental comparative analysis between our method and the SFT+RL method under the same settings. In addition, we clearly elaborated on the contributions of our method, the definition of GF, and the cost related to our method. All of the above have been improved in the revised paper.
>
> Your suggestions are of great significance for improving our work. We would be very grateful and look forward to receiving your feedback on our responses during the discussion.
>
> Thank you for your time and understanding.
>
> Sincerely,
>
> Authors

---

> > ### Comment · Reviewer_bc3X · 2025-11-27
> >
> > Thanks to the authors for the rebuttal. However, the key concerns remain unresolved:
> >
> > - **Role of the zero-model:** The proposed “GRPO-zero” paradigm differs from DeepSeek-R1-Zero, which applies GRPO directly on a *base* model to induce reasoning behaviors through pure RL. In contrast, your method applies GRPO on Qwen2.5-VL-7B-Instruct (an already instruct-tuned model). Naturally, the resulting “zero” model exhibits higher format and answer accuracy—and therefore higher “data utilization”—but this doesn't indicate that GRPO-zero provides any substantive improvement in the model’s reasoning generalization.
> >
> >   Since the zero-model has already done RL finetuning, it is expected that performing DPO (with additional SFT loss) on its outputs leads to better cold-start initialization than using responses from the base model. However, the cold-start advantage does not imply better final performance after the final GRPO training. A fairer comparison would run the final GRPO stage on (i) a DPO-initialized model distilled from your GRPO-zero responses and (ii) a DPO-initialized model distilled from Qwen2.5-VL-7B-Instruct responses, to directly assess whether GRPO-zero provides any actual warm-up benefit for the final RL training.
> >
> > -  **Efficiency**: The main computational cost is not the amount of SFT/DPO cold start data, but rather the GRPO stages themselves. GRPO requires multiple on-policy autoregressive rollouts per step, making it much more expensive than offline objectives such as SFT or DPO, which require only a single forward pass per sample.
> >
> >    As GRPO is the stage that requires the most compute and usually contributes most to the performance gain in reasoning, its cost should be carefully analyzed; however, this is not adequately discussed in the paper. (The proposed methods requires two stages of GRPO training, while other papers only have a single GRPO stage.)
> >
> > - **Missing important baseline**: To better validate the proposed cold-start strategy, it would be important to run GRPO finetuning (the proposed “zero” stage) for the same total number of RL steps as the sum of GRPO-zero and the final GRPO stage. Only then can we isolate whether DPO cold-start with GRPO-zero provides an actual benefit, rather than simply adding more RL compute.

---

> > > ### Author Response · Authors · 2025-11-27
> > > **Response to Follow-up Questions (1 / 2)**
> > >
> > > Thank you very much for your constructive feedback!
> > >
> > > **Role of the zero-model**
> > >
> > > > "but this doesn't indicate that GRPO-zero provides any substantive improvement in the model’s reasoning generalization."
> > >
> > > In fact, as summarized in the Related Work section of the revised paper, most excellent current works improve the reasoning ability of Instruct models through direct RL on them. Some of these works, such as [1-4], perform RL on the Instruct models of the Qwen2.5 series. It should be noted that this series has not open-sourced any size of the unfinetuned base model. The improvement in reasoning ability on Pass@1 can be observed from the accuracy performance of the model after RL, model response analysis, and model case studies.
> > >
> > > I would like to provide a clearer explanation of our motivation for using GRPO-zero. The main reason we chose GRPO-zero for data generation is that it is closer to the base model and achieves better performance. Specifically, GRPO-zero, which is directly trained via RL, exhibits a smaller shift in the token probability distribution compared with the original model trained via SFT[5] or other models series. Thus, distilling from GRPO-zero also aligns with the concept of self-distillation.
> > >
> > > This superior performance is reflected in two key aspects. On the one hand, it is reflected in the data utilization rate we discussed in Reply 1. On the other hand, it helps improve the quality of the chosen responses. This improvement includes not only its stronger adherence to formatting requirements but also its improved reasoning capabilities after RL training (This can be verified by higher answer accuracy, an increased number of reasoning-related words, and findings from existing works).
> > >
> > > > "However, the cold-start advantage does not imply better final performance after the final GRPO training. A fairer comparison would run the final GRPO stage on ..."
> > >
> > > We fully agree with this comment. In fact, we have already supplemented this completely fair experiment in our first reply to you and in Table 4 of the revised paper. I would like to provide it again in this reply:
> > >
> > > | Model                     | MegaBench     | MMMU          | MathVista     | MathVision    | MathVerse     | Eval          |
> > > | ------------------------- | ------------- | ------------- | ------------- | ------------- | ------------- | ------------- |
> > > | Qwen2.5-7b-instruct       | 35.07         | 54.2          | 63.70         | 25.40         | 38.20         | 43.31         |
> > > | distillation by Qwen2.5-7B     | 35.37 / 37.92 | 53.11 / 56.11 | 67.9 / 74.4   | 25.55 / 28.68 | 43.40 / 46.82 | 45.07 / 48.79 |
> > > | distillation by GRPO-Zero | 37.52 / 39.17 | 54.89 / 56.78 | 72.00 / 75.90 | 25.75 / 29.50 | 46.19 / 48.73 | 47.27 / 50.02 |
> > >
> > > It can be seen from the above experimental results that if the Qwen2.5-VL-7B-Instruct model is directly used for distillation, this process is not only more difficult (due to inefficient data utilization), but the final results also decrease due to the quality of the data.
> > >
> > > We acknowledge that our explanation of this motivation was insufficiently clear. We will provide a more explicit explanation of the motivation for using GRPO-zero in the revised manuscript.
> > >
> > >
> > >
> > > [1] Meng F, Du L, Liu Z, et al. Mm-eureka: Exploring the frontiers of multimodal reasoning with rule-based reinforcement learning[J]. arXiv preprint arXiv:2503.07365, 2025.
> > >
> > > [2] Peng Y, Zhang G, Zhang M, et al. Lmm-r1: Empowering 3b lmms with strong reasoning abilities through two-stage rule-based rl[J]. arXiv preprint arXiv:2503.07536, 2025.
> > >
> > > [3] Ma Y, Du L, Shen X, et al. One RL to See Them All: Visual Triple Unified Reinforcement Learning[J]. arXiv preprint arXiv:2505.18129, 2025.
> > >
> > > [4] Shen H, Liu P, Li J, et al. Vlm-r1: A stable and generalizable r1-style large vision-language model[J]. arXiv preprint arXiv:2504.07615, 2025.
> > >
> > > [5] Fu Y, Chen T, Chai J, et al. SRFT: A Single-Stage Method with Supervised and Reinforcement Fine-Tuning for Reasoning[J]. arXiv preprint arXiv:2506.19767, 2025.

---

> ### Author Response · Authors · 2025-11-27
> **Response to Follow-up Questions (2 / 2)**
>
> **Efficiency & Missing important baseline**
>
> We selected the 400th step (with a batch size of 128; more training parameters can be found in Appendix F of the revised paper) for both the cold-start phase and the final GRPO phase of the model. This selection takes into account the model's convergence and training efficiency. In fact, we trained 800 steps in both the GRPO-zero phase and the final GRPO phase, and we have included the performance of the model from step 0 to step 800 in the final GRPO phase in Figure 4 of the revised paper. Due to the constraints of the model size, if we only train GRPO on the model, the performance improvement of the model becomes very limited after 400 steps.
>
> The performance of the model GRPO-zero (GRPO-400 steps), our model Ours (DPO+GRPO 400 steps), and the model trained solely with 800 steps of GRPO (which can be understood as GRPO-zero-800steps, but since it is not the GRPO-zero we used for data generation, we still refer to it as GRPO-800 steps) on various benchmarks is as follows:
>
> | Model                          | MathVista | MathVision | MathVerse | MegaBench | MMMU      | Eval      |
> | ------------------------------ | --------- | ---------- | --------- | --------- | --------- | --------- |
> | Qwen2.5-7B-Instruct            | 63.70     | 25.40      | 38.20     | 35.07     | 54.2      | 43.31     |
> | GRPO-zero （GRPO-400 steps）   | 72.9      | 26.88      | 47.33     | 37.96     | 54.3      | 47.87     |
> | GRPO-800 steps                 | 74.1      | 27.47      | 47.71     | 37.92     | 54.3     | 48.3     |
> | **Ours（DPO+GRPO 400 steps）** | **75.90** | **29.50**  | **48.73** | **39.17** | **56.78** | **50.02** |
>
> As can be seen from the table, our scheme still has a significant advantage compared with the sum of directly training GRPO-zero and the final GRPO steps.
>
> Regarding time consumption and training efficiency, we have provided a clearer explanation of time consumption in the Appendix F. Our GRPO phase takes approximately 12 hours on 32 NVIDIA H800 (80G) GPUs. If we combine the GRPO-zero phase, the cold-start phase, and the final GRPO phase, the total time consumption is approximately 25 hours. Although we have two GRPO phases, we believe the total training token consumption is acceptable.
>
> By contrast, Vision-R1-7B[6] was trained for 10 hours using 32 NVIDIA H800 80G GPUs during the cold-start phase, and for approximately 2 days using 64 NVIDIA H800 80G GPUs during the RL phase.
>
> [6] Huang W, Jia B, Zhai Z, et al. Vision-r1: Incentivizing reasoning capability in multimodal large language models[J]. arXiv preprint arXiv:2503.06749, 2025.
>
> ---
>
> If there are any additional points or feedback regarding our new results and explanations that you'd like us to consider, please let us know. Your insights are invaluable to us, and we are eager to engage in further discussion to resolve these issues.
>
> Thank you for your time and effort in reviewing our paper.

---

### Official Review · Reviewer_T4Pk · 2025-11-01

**Soundness:** 4
**Presentation:** 3
**Contribution:** 3
**Rating:** 6
**Confidence:** 3

**Summary:**

This paper introduces SPECS, a Self-Distilled Preference-based Cold-Start framework that rethinks initialization for reinforcement-learning-based multimodal large language models (MLLMs). The authors demonstrate that standard supervised fine-tuning (SFT) couples reasoning content and output format, resulting in overfitting and poor out-of-distribution (OOD) generalization.
SPECS decouples learning objectives through three stages:
1. Self-Distillation – Generates preference pairs focused on output format without large teacher models.
2. DPO-based Pre-Alignment – Applies Direct Preference Optimization with a hybrid (DPO + SFT) loss for stable format learning.
3. GRPO Fine-Tuning – Performs final reinforcement learning with verifiable rewards for deep reasoning.
A new Generalization Factor (GF) metric measures cold-start generalization and correlates strongly (R² ≈ 0.86) with final RL performance.
Across MEGA-Bench, MMMU, MathVista, MathVision, and MathVerse, SPECS achieves consistent gains over strong baselines (e.g., VL-Rethinker, MM-Eureka), improving +4.1 % on MEGA-Bench and +12.2 % on MathVista. Ablations confirm that self-distillation and decoupled data outperform teacher-based and coupled variants.

**Strengths:**

- The paper tackles a pressing issue in multimodal RL pipelines—how to design an effective cold-start phase for MLLM-r1 models—complementing recent efforts such as Vision-R1 and Advancing Multimodal Reasoning via RL with Cold Start.
- The self-distilled preference data strategy avoids dependence on large teacher models while maintaining strong alignment quality. The decoupled DPO training elegantly separates surface-form learning from reasoning acquisition.
- The proposed Generalization Factor (GF) offers a measurable and interpretable indicator of generalization potential, empirically shown to correlate with final RL performance.
-  Evaluations across multiple reasoning benchmarks, together with well-designed ablations (self-distillation vs. teacher-distillation; decoupled vs. coupled data), substantiate the paper’s claims.
- The writing is structured and transparent, with detailed hyperparameters, dataset composition, and implementation frameworks (MM-EUREKA, LlamaFactory) that support reproducibility.

**Weaknesses:**

- The related-work section omits recent concurrent methods such as Vision-R1 (Huang et al., 2025), AdaViP (Lu et al., 2025), Chain-of-Focus (2025), and DeepEyes (2025), which would situate SPECS more clearly within the multimodal RL ecosystem.
- Current evaluations emphasize mathematical reasoning; inclusion of more general-domain benchmarks (e.g., ScienceQA, MMBench) would better demonstrate scalability.
- The GF analysis is confined to one model family (QwenVL). Extending the metric across other backbones (e.g., InternVL, Kimi-VL) would confirm its generality.
-  Dependence on GPT-4o and Gemini-2.5 for preference filtration introduces potential bias; multi-judge or calibration procedures could strengthen reliability.
- While the framework improves RL convergence, the additional computational cost of the DPO stage is not explicitly quantified.

**Questions:**

1. How does SPECS empirically or conceptually differ from Vision-R1 and AdaViP, which also integrate RL and preference-based training?
2. Can the proposed GF metric predict RL success across different architectures (InternVL, Kimi-VL)?
3. What is the size and diversity of the self-distilled preference dataset, and how does scaling influence results?
4. How were biases in GPT-4o/Gemini evaluations mitigated (e.g., standardized prompts or aggregation)?
5. Could SPECS generalize to text-only or perception-heavy domains (e.g., R1-VL, Chain-of-Focus models)?

---

> ### Author Response · Authors · 2025-11-18
> **Reply (Part 1 / 2)**
>
> Thank you very much for your valuable comments and suggestions. Below we  address each of them in detail.
>
> **W1 & Q1: Supplementary to recent work.**
>
> Both Vision-r1 and SPECS aim to enhance the reasoning ability of MLLMs. Vision-R1 adopts the conventional approach, SFT cold start + RL training. It uses DeepSeek-R1 to construct a 200K multimodal COT dataset and then employs GRPO to improve the model's reasoning ability. We focus on the cold start phase. In contrast, we only use 9K self-distilled preference data and adopt the DPO cold start scheme, which outperforms Vision-r1 on multiple benchmarks.
>
> The main goal of AdaViP is to enhance the model's sensitivity to visual details. This work constructs "preferred image - rejected image" pairs by precisely removing key visual elements from images, thereby improving the sensitivity of MLLMs to visual details. It only uses DPO and does not conduct experiments on cold start and subsequent RL, showing significant differences in both training objectives and specific methods.
>
> In the Related Work section of our paper, we introduced two relevant research topics: "RL for Multimodal Reasoning" and "Cold-Start Strategies and Data Generation". We will modify Related Work to provide a more comprehensive  account in the  revised paper.
>
> **W2: Regarding evaluation benchmarks.**
>
>  The reasons why we focus on assessments related to mathematical reasoning are: 1) Mathematical reasoning is an evaluation metric used in most related works, making it easier for us to compare with them. 2) Mathematical reasoning is objective and deterministic, and mathematical reasoning ability is also the key to transfer to other complex fields.  In the original paper, besides evaluating mathematical reasoning , we also tested general reasoning benchmarks such as Megabench and MMMU. In addition, we also provide the evaluation of our model on ScienceQA and MMBench. The following are the general reasoning results:
>
> | **Model**           | **Megabench** | **MMMU** | **ScienceQA** | **MMBench** |
> | ------------------- | ------------- | -------- | ------------- | ----------- |
> | Qwen2.5-7b-instruct | 35.1          | 54.2     | 85.0          | 82.4        |
> | Ours                | 39.2          | 56.8     | 91.6          | 82.3        |
> | $\Delta$            | +4.1          | +2.6     | +6.6          | -0.1        |
>
> Among them, there is no significant improvement in MMBench. After analysis, we believe that MMBench mainly tests the model's knowledge ability. Compared with other benchmarks, it places less emphasis on logical reasoning ability, while our main training objective is to enhance the model's logical reasoning ability.
>
> **W3 & Q2: The GF analysis.**
>
> To further demonstrate the impact of DPO and SFT training on the generalization ability of models, in addition to the Qwen2.5VL-7B-Instruct model, we conducted tests on InternVL3-8B-Instruct. The conclusions are consistent with those we obtained on Qwen2.5VL-7B-Instruct: as training progresses, SFT impairs the model's performance on OOD tasks and affects the model's GF. The table below shows the test results of the InternVL3-8B-Instruct model at the same training steps using the same training and evaluation parameters.：
>
> | model                 | OOD Performance gains | ID Performance gains | Generalization Factor |
> | ------------------------- | ------------------------- | ------------------------ | ------------------------- |
> | InternVL3-8B-Instruct-SFT | 10.43                     | 21.00                    | 11.60                     |
> | InternVL3-8B-Instruct-DPO | **14.05**                 | **21.76**                | **15.12**                 |
>
> **W4 & Q4: Regarding data filtering**
>
> In fact, the main purpose of using Gemini-2.5 flash for data filtering is to screen out data where the answers are consistent with the reasoning, avoiding situations where the grpo-zero reasoning process is inconsistent with the answers, which could affect training, such as the following case:
>
> > \<think\>To determine the total number of dogs in the picture, we need to count each dog individually. From the image, we can see that there are **4** dogs. \</think\> \n \<answer\> The answer is \boxed{**3**} \</answer\>
>
> This evaluation process is simple and objective, with no significant errors, and will not introduce bias into data filtering. Through a certain batch of manual verification, it has been confirmed that the filtering error is within an acceptable range. If multi-judge is used, we believe it will bring unnecessary additional costs to data choosing. However, we will further investigate the causes of such situations to fundamentally prevent them from occurring.

---

> ### Author Response · Authors · 2025-11-18
> **Reply (Part 2 / 2)**
>
> **W5: Analysis of additional computational cost in the DPO phase**
>
> We understand the authors' concern about the additional computational cost of using DPO compared to SFT. However, through theoretical analysis and experimental evidence, unlike other RLHF methods that require a Reward Model to calculate preferences, the design of DPO allows its training speed to be not significantly different from that of SFT. We have counted the computing time for DPO and SFT training under the same training parameters (including steps, batch size, data length, max new tokens, etc.), as shown in the following table:
>
> |method | train samples/s | train steps/s | total flos | run time (s) |
> | ------------------- | ---------------------------- | -------------------------- | -------------- | --------------- |
> | SFT                 | 1.035                        | 0.129                      |  $2.82\times 10^{17}$            | 8848            |
> | DPO                 | 0.979                        | 0.123                      | $6.22\times 10^{17}$             | 9346            |
> | $\Delta$          | -0.056                       | -0.006                     | $+3.4\times 10^{17}$             | +498            |
>
> As it can be seen from the table, in actual computation, the computational cost between SFT and DPO is not significantly different. For the computation of 9,000 pieces of data, the total computation time only increased by approximately 8.3 minutes.
>
> **Q3：Regarding the size and diversity of self-distillation data**
>
> During the training process, we used 9K DPO cold-start preference data sampled from Orsta47K and virl39K, which covers mathematical reasoning and general reasoning. Generally speaking, cold-start training conforms to scaling laws. As can be observed from Figure 5 in the original paper, during the RL phase, the model size increases steadily in line with the data size.. we plan to  further investigation   on the scaling influence of self-distillation preference datasets  in next step.
>
> **Q5: Regarding domain expansion**
>
> As mentioned in the Conclusion section of our paper, our current work focuses on the field of multimodal reasoning. However, whether this training paradigm can be extended to the pure text domain and visual perception domain requires further research in terms of data construction, training analysis, and other aspects. We believe that theoretically, the conclusion that using DPO instead of SFT for cold start to improve the model's generalization ability during the cold start phase can make SPECS inspire related work in other fields as well.
>
> Thank you again for your valuable suggestions. We will refine  the main text and appendix of the revised paper accordingly.

---

> ### Author Response · Authors · 2025-11-27
> **Gentle Reminder: Feedback on Response**
>
> Dear reviewer T4Pk：
>
> We are truly grateful for your constructive comments on our submission.
>
> In response to your review, we have added extensive experiments to clarify and answer your concerns. 1) We have provided a more comprehensive introduction to our recent work and updated it in the Related Works section of the revised article. 2) We have conducted a more comprehensive evaluation on the general reasoning benchmarks you mentioned (ScienceQA and MMBench), to demonstrate our scalability. 3) We have extended the analysis of GF to the InternVL series to prove its generality. 4) We have made a more comprehensive comparison of the additional computational overhead in the DPO stage with that in the SFT stage. The above parts of the content have been improved in our revised paper.
>
> Your suggestions are of great significance for improving our work. We would be very grateful and look forward to receiving your feedback on our responses during the discussion.
>
> Thank you for your time and understanding.
>
> Sincerely,
>
> Authors

---

### Author Response · Authors · 2025-11-18
**List of Revisions**

We sincerely thank you for your thoughtful and constructive feedback. We attach great importance to every comment and suggestion. According to the reviewers' suggestions, we have submitted a revised version of the paper. This revised paper mainly supplements some ablation experiments and refines some expressions. The main revisions are as follows:

1. In **Section 2.1** of the revised paper, we have expressed the calculation formula and theoretical basis of GF in a more simplified manner.
2. In **Section 4.3** of the revised paper, we have added an analysis of the necessity of GRPO-zero in experiments.
3. In **Section 4.4** of the revised paper, we have added a comprehensive comparison between SFT+GRPO and DPO+GRPO, as well as an analysis of training stability.
4. In **Section 5** of the revised paper, we have introduced the Related Works in more detail.
5. In **Appendix E** of the revised paper, we have added an analysis of using Gemini for chosen response filtration.
5. In **Appendix F** of the revised paper, we have added specific experimental details and time costs.

In addition, some other detailed contents has been improved. We appreciate the valuable comments from each reviewer. Next, we will reply to each reviewer's comments one by one within a few hours, and look forward to further discussions.

---

### Author Response · Authors · 2025-12-01
**Summary of the Discussion Phase and Paper Revisions**

Dear Area Chair,

We sincerely thank you and the reviewers for the time and effort devoted to assessing our submission. Below we briefly summarize the discussion phase and the corresponding revisions.

First, we appreciate that all four reviewers acknowledge the contribution and novelty of this work. They generally consider that this paper has a clear structure and is easy to understand (*8EWE, bc3X, T4Pk*). The method proposed in this work is intuitive and effective, with significant performance improvements in various benchmark tests (*T4Pk, bc3X, hLrc, 8EWE*). The proposed technical insights, such as format decoupling and self-distillation strategies, are reasonably motivated and supported by sufficient effective evidence (*T4Pk, bc3X, hLrc*). The definition of GF and the analysis of SFT and DPO are interesting, practical, and interpretable (*T4Pk, hLrc, 8EWE*).

Second, in response to the weaknesses and questions raised during review and discussion, we have made the following clarifications and additions:

**Revisions to the paper:**

**1. Regarding the use of GRPO-zero (*hLrc, bc3X*):**

We analyzed the necessity of using GRPO-zero and conducted ablation experiments where GRPO-zero was not used, and instead, data distilled from the base model was employed. We analyzed the acceptable additional consumption brought by GRPO-zero. All experimental results are consistent with our conclusions. The relevant content has been updated in Section 4.3 of the revised paper.

**2. Regarding the effectiveness, stability, and additional cost of DPO cold start (*hLrc, bc3X, T4Pk*)**

We comprehensively compared the differences between DPO and SFT using self-distilled cold start data, and verified the effectiveness of our method through multiple comparative validations. We analyzed the potential reasons why DPO incurs minimal additional overhead compared to SFT and why DPO brings stable training. The relevant content has been updated in Section 4.4 of the revised paper.

**3. Regarding the use of Gemini evaluation (*hLrc, T4Pk, bc3X*)**

We have more clearly elaborated on the purpose of using Gemini for data filtering and the necessity of such filtering. Through ablation experiments without data filtering and ablation experiments with Gemini distillation, we have supplementarily proven the rationality and effectiveness of our method. Through the analysis of data cases and the statistics of API consumption, we have further illustrated the model replaceability and low consumption of Gemini filtering. The relevant content has been updated in Appendix E of the paper.

**4. Regarding the refinement of the paper's expression (*bc3X, hLrc, T4Pk*)**

We have more clearly elaborated on the definition of GF; supplemented and improved recent parallel works; added more comprehensive training detail parameters; the relevant content has been updated in Section 2.1, Section 5, and Appendix F of the paper.

**Discussions with reviewers:**

1. Response to *bc3X*'s Follow-up Questions (2 / 2)

We presented comparative experiments where we increased the final RL training volume without using GRPO-zero, and the experimental results demonstrated the effectiveness of our method.

2. Reply to *hLrc* (Part 2 / 2)

We provided the reason analysis and significance analysis of the ablation experiments on coupled data.

3. Reply to *T4Pk* (Part 1 / 2)

The extended tests on general domain benchmarks and the analysis of other model series on GF all yielded results that were in line with expectations.

4. Reply to *bc3X* (Part 1 / 3)

Regarding the elaboration of the contributions of this paper.

---

In addition, during the discussion phase, reviewer hLrc indicated that our response had well addressed the concerns and **raised the score from 4 to 6** on 21 Nov 2025, at 11:04 (AOE Time).

Thank you for your time and we hope this summary is helpful for your decision.

Sincerely, Authors

---

### Meta-Review · Area_Chair_eyNU · 2026-01-05

**Summary:**

The reviewers generally recognized the effectiveness of the SPECS framework, specifically the concept of using self-distilled, preference-based learning to cold-start Vision Language Models before Reinforcement Learning.
The initial concerns focused heavily on the fairness of baselines, the justification for the multi-stage pipeline (specifically the GRPO-zero stage), and the generalizability of the proposed metric (Generalization Factor) and the method itself beyond mathematical reasoning.

**Reviewer Concerns:**

1. Reviewer bc3X strongly challenged whether the gains were due to the method or simply running RL for longer (since the method involves two GRPO stages). The authors effectively addressed this in the discussion by providing a comparison between SPECS (DPO + GRPO) and a Pure GRPO baseline with equivalent total steps (800 steps). SPECS outperformed the baseline, proving the value of the cold-start phase.
2. The definition of the Generalization Factor (GF). The authors clarified that GF is essentially an F-beta score balancing in-distribution (ID) and out-of-distribution (OOD) gains, and extended its evaluation to other model families (InternVL), satisfying T4Pk.
3. The authors provided concrete cost breakdowns (e.g., $15 for Gemini filtering, ~25 GPU hours total), arguing that the cost is comparable to or lower than concurrent baselines like Vision-R1.
4.  The method requires a multi-step pipeline (GRPO-zero -> Self-Distillation -> GRPO). Reviewer bc3X might still view this as "over-engineered" compared to a simpler pipeline, even if the empirical results are superior.

**Reviewer Scores:**

Reviewer T4Pk (6) and Reviewer 8EWE (8) were already positive. Reviewer hLrc (4->6) was satisfied with the explanation of the GRPO-zero motivation and the training stability analysis.

Reviewer bc3X (2) has the concerns of (1) technical contribution, (2) the necessity of the zero-mode, (3) the definition of Generalization Factor, (4) cost analysis, (5) the baseline of SFT cold start + RL. In the final discussion round, the authors provided the exact table bc3X requested: a comparison of GRPO-800 steps vs. SPECS (DPO+GRPO 400 steps). The data showed SPECS was superior, directly refuting the reviewer's main technical objection.

---

### Decision · Program_Chairs · 2026-01-26

Accept (Poster)